# Reversible photoregulation of cell-cell adhesions with opto-E-cadherin

Brice Nzigou Mombo [1], Brent M. Bijonowski[1], Christopher A. Raab[1], Stephan Niland [1], Katrin Brockhaus[1], Marc Müller[1], Johannes A. Eble [1] & Seraphine V. Wegner [1] ✉

E-cadherin-based cell-cell adhesions are dynamically and locally regulated in many essential processes, including embryogenesis, wound healing and tissue organization, with dysregulation manifesting as tumorigenesis and metastasis. However, the lack of tools that would provide control of the high spatio-temporal precision observed with E-cadherin adhesions hampers investigation of the underlying mechanisms. Here, we present an optogenetic tool, opto-E-cadherin, that allows reversible control of E-cadherin-mediated cell-cell adhesions with blue light. With opto-E-cadherin, functionally essential calcium binding is photoregulated such that cells expressing opto-E-cadherin at their surface adhere to each other in the dark but not upon illumination. Consequently, opto-E-cadherin provides remote control over multicellular aggregation, E-cadherin-associated intracellular signalling and F-actin organization in 2D and 3D cell cultures. Opto-E-cadherin also allows switching of multicellular behaviour between single and collective cell migration, as well as of cell invasiveness in vitro and in vivo. Overall, opto-E-cadherin is a powerful optogenetic tool capable of controlling cell-cell adhesions at the molecular, cellular and behavioural level that opens up perspectives for the study of dynamics and spatiotemporal control of E-cadherin in biological processes.

Cell–cell adhesions are dynamically regulated in biology to fundamentally control cell behaviour in changing contexts[1]. They are essential during many processes, such as embryonic development, wound healing, collective cell migration and maintenance of tissue integrity, with misregulation resulting in pathologies such as tumorigenesis and cancer metastasis[2]. Epithelial-cadherin (E-cadherin) is the adhesion molecule that underlies these cell–cell adhesions, where the extracellular domains of E-cadherins on neighbouring cells bind to each other[3]. As a result, intracellular proteins are recruited to the cytoplasmic tail of E-cadherin and connect the adhesions to the actin cytoskeleton and transcriptional regulation[2,4,5]. These adhesions are dynamically and locally regulated in the epithelial-mesenchymal transition (EMT), where the loss of E-cadherin expression and epithelial characteristics results in a mesenchymal cell phenotype, which may revert to epithelial cells during the mesenchymal-epithelial transition[6–8]. For example, EMT plays a central role in gastrulation, tissue morphogenesis in development and wound healing in adults[7], whereas the loss, reduction or dysfunction of E-cadherin is systematically seen in most aggressive and undifferentiated carcinomas of the mammary gland and other epithelial tissues[9,10].

Methods to dynamically turn E-cadherin-mediated cell–cell adhesions on or off with high spatiotemporal precision are of great interest for investigating the underlying mechanisms. Widely used approaches are genetic manipulations that alter the expression levels of cadherins[11], their inhibition with antibodies[12] or the removal of $Ca^{2+}$ ions[13]. Alternatively, the introduction of chemical reactive groups allows cells to be brought together by using click chemistry[14], DNA nanotechnology[15,16] and other unnatural adhesion molecules[17].

[1]Institute of Physiological Chemistry and Pathobiochemistry, University of Münster, Waldeyerstraße 15, 48149 Münster, Germany. ✉e-mail: wegnerse@uni-muenster.de

Nonetheless, degradation of these non-genetically coded modifications limits their use and most importantly, none of these methods provide the desired high spatiotemporal control. Such spatiotemporal control is possible, however, when using light-controlled cell–cell adhesions[18–20]. The current state of the art involves an optochemical[21], and optogenetic tools that allows light-induced dissociation of adherens junctions in through photocleavable linkers[22], photoswitchable small molecules introduced through metabolic labelling[23], and artificial photoswitchable surface proteins that mediate artificial cell–cell interactions[24–27]. Yet, these tools are limited by the need for light-responsive chemicals, lack of reversibility, use of UV light or the fact that they do not link to E-cadherin-associated cell signalling.

Here, we present a one-component optogenetic tool called opto-E-cadherin (opto-E-cad) that provides reversible control over E-cadherin-based cell–cell adhesions with high spatiotemporal precision. Opto-E-cad mediates cell–cell adhesions in the dark, which can be disassembled upon blue light illumination and assembled again once illumination is stopped. We demonstrate that opto-E-cad allows spatiotemporal photoregulation of cell–cell adhesions in 2D and 3D cell culture systems with consequences for cell signalling, cell migration and invasion behaviour in vitro and in vivo.

## Results

### Design of opto-E-cad

In designing a photoswitchable E-cadherin, opto-E-cad, we engineered functionally essential $Ca^{2+}$ binding sites in E-cadherin such that the $Ca^{2+}$ complexation became light switchable (Fig. 1a). In particular, we focused on the calcium binding site between the first and second extracellular domains (EC1 and EC2), which directly interact on opposing cells when cell–cell adhesions form. After careful analysis of the crystal structure (PDB: 2O72, 1FF5)[28,29], and on the basis of previous work[30], we identified a connecting loop (D134–I146) between two beta sheets in EC2 as an appropriate point to insert the blue light-switchable LOV2 domain (Supplementary Fig. 1a). Part of this loop are the side chains of D134 and D136 and the carbonyl oxygen of N143, which are ligands for two of the three $Ca^{2+}$ ions between EC1 and EC2. The LOV2 domain of *Avena sativa* phototropin 1 (AsLOV2, 404–542) inserted before D134 has a well-folded Per-Arnt-Sim (PAS) domain and a flavin

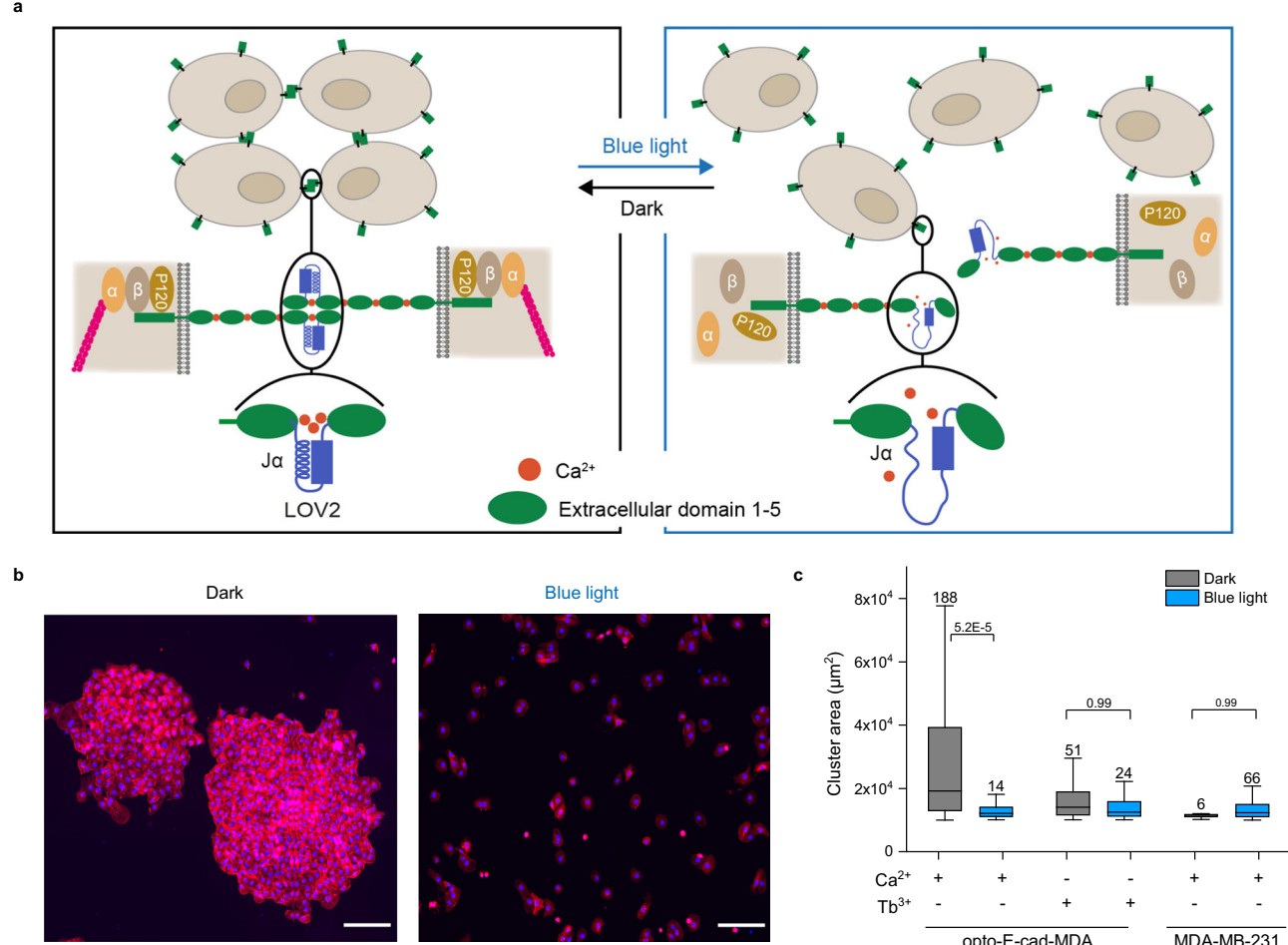

**Fig. 1 | Design and validation of the opto-E-cad. a** Schematic representation of the opto-E-cad. Cells that express opto-E-cad on their surfaces form cell–cell adhesions in the dark but don't blue light. In the design of opto-E-cad, the LOV2 domain is inserted between the first and second extracellular domains E-cadherin in proximity to one of the calcium binding sites. In the dark, the Jα-helix of the LOV2 remains folded such that the $Ca^{2+}$ ions can bind and the E-cadherins on neighbouring cells interact. Under blue light, the Jα-helix unfolds such that the $Ca^{2+}$ ions cannot bind and the E-cadherin interactions are lost. **b** Fluorescence microscopy images of opto-E-cad-MDA cells on glass surfaces after 4 h in the dark or under blue light. Actin shown in red, nuclei shown in blue. In the dark the cells grow in large clusters but under blue light grow as single cells. Scale bar is 200 μm. **c** Quantification of cluster areas, from left to right $n = 6, 4, 5, 4, 4, 3$ independent experiments. The average number of clusters detected in each experiment is shown above the box plot. Boxplots show median, 25th, and 75th percentile. The lower and upper boundaries of whiskers indicate the minima and maxima, respectively. All comparisons are performed using Fisher's One Way ANOVA test and $p < 0.05$ was treated as the significance threshold. Source data are provided as a Source Data file.

mononucleotide (FMN) chromophore cofactor[31]. Upon blue light illumination, a cysteine at the core of the AsLOV2 domain reacts with FMN, the protein undergoes conformational changes, the C-terminal Jα-helix unfolds and consequently affects the activity of adjacent domains. This photoreaction and the conformational changes are reversible in the dark and have been used to design reversible single-component optogenetic tools[32]. In our design of the opto-E-cad, we hypothesized that the inserted LOV2 domain would not affect $Ca^{2+}$ binding and the cell−cell adhesions in the dark, because in the dark state, the N- and C-termini of AsLOV2 are spatially close to each other. Upon blue light illumination, the C-terminal Jα-helix would unfold and the directly linked D134 and D136 would become unstructured, which should diminish $Ca^{2+}$ affinity and, hence, disturb the E-cadherin-mediated cell−cell adhesions. When inserting the LOV2 domain at the chosen position, we further paid attention not to sterically block homophilic cis- and trans-interactions between E-cadherins by modelling the opto-E-cad structure with AlphaFold and aligning it with the crystal structures of the E-cadherin dimers (Supplementary Fig. 1b, c). In both cases, the LOV2 domain is oriented in a way that does not affect the interaction interface.

## Opto-E-cad adhesions depend on light and calcium

First, we investigated whether opto-E-cad performs as envisioned in the design. To this end, we constructed it starting from the human E-cadherin with a C-terminal green fluorescent protein (GFP) tag in a mammalian expression vector and transfected it into the breast cancer cell line MDA-MB-231, which does not express type 1 cadherins and has only weak cell−cell adhesions[33]. Using the selectable geneticin marker and the GFP label on the protein, we established a stable monoclonal cell line, opto-E-cad-MDA, which expresses opto-E-cad on its plasma membrane (Supplementary Fig. 2a, b).

When we seeded the opto-E-cad-MDA at sub-confluent densities (8600 cells/cm$^2$) in the presence of $Ca^{2+}$ (1.8 mM typical concentration in the medium) on glass substrates in the dark or under blue light, the cells behaved differently. In the dark, the cells grew in groups that were visible with the nuclei staining (shown in blue) and the actin cytoskeleton staining (shown in red), a sign of strong cell−cell adhesions (Fig. 1b, Supplementary Fig. 3b). In contrast, under blue light, the cells remained single cells and were evenly distributed over the glass slide with few contacts between them. In fact, the opto-E-cad-MDA cells under blue light resembled the parent MDA-MB-231 cells in their morphology and lacked the cell−cell adhesions. We added terbium ions ($Tb^{3+}$) to the opto-E-cad-MDA cells as a negative control, which compete out $Ca^{2+}$ ions in cadherins and disturb cell−cell adhesions[34]. Unlike in the presence of $Ca^{2+}$ ions, the cells grew as single cells in the dark in the presence of $Tb^{3+}$ (Supplementary Fig. 3b), which shows the importance of $Ca^{2+}$ binding for cell−cell adhesions. When we quantified cell clustering in these 2D cultures, we observed that the opto-E-cad-MDA cells formed cell clusters with a larger area in the dark than that under blue light illumination (Fig. 1c). Moreover, clustering under blue light for opto-E-cad-MDA cells was similar to that of negative controls with $Tb^{3+}$ and of parent MDA-MB-231 cells, and the blue light illumination did not alter clustering in these negative controls. These differences in cluster sizes did not arise from differences in seeding density, because all samples had equal numbers of cells as determined from the nuclei stain (Supplementary Fig. 3a). In this and in all other experimental set-ups described below, we observed no toxicity of the blue light (Supplementary Fig. 4a−c). These results demonstrated that the proposed opto-E-cad design functions as envisioned and is an efficient optogenetic tool to alter $Ca^{2+}$ binding to E-cadherin and cell−cell adhesions by using blue light.

Next, to characterize the kinetics of opto-E-cad, we used a cell aggregation assay in suspension culture. This assay allows the study of cell−cell adhesions independent of adhesions to any substrate, as well as observation of changes in cell−cell adhesions in shorter time scales.

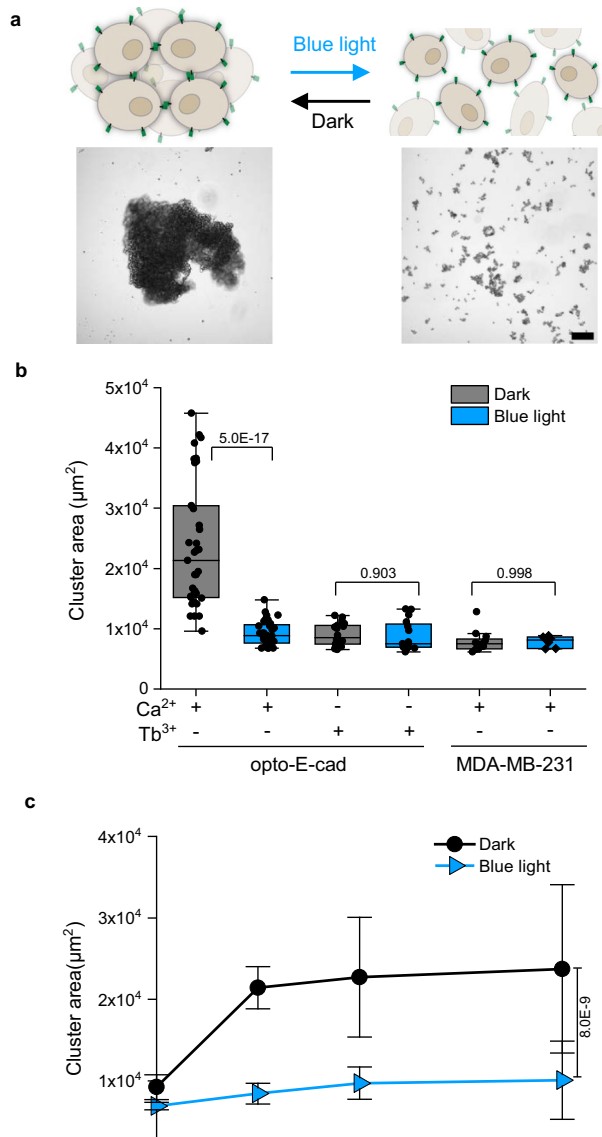

**Fig. 2 | Light and calcium response of opto-E-cad in suspension culture. a** Schematic representation (upper panels) and bright field images (lower panels) of opto-E-cad-MDA cells ($5 \times 10^4$ cells/ml) in suspension culture in the dark and under blue light incubated for 2 h at 30 rpm. In the dark, the cells form large aggregates but under blue light illumination remain as single cells. Scale bar is 50 μm. **b** Quantification of average cluster area, from left to right $n = 35, 32, 18, 16, 12, 7$ samples examined over at least 3 independent experiments. Boxplots show median, 25th, and 75th percentile. The lower and upper boundaries of whiskers indicate the minima and maxima, respectively. All comparisons are performed using Fisher's One Way ANOVA test and $p < 0.05$ was treated as the significance threshold. **c** Quantification of average cell cluster area as a function of $Ca^{2+}$ (two-tailed $t$-test, $n = 6$). Data are represented as mean values ± SD. Source data are provided as a Source Data file.

When we incubated the opto-E-cad-MDA cells on a 3D orbital shaker at 30 rpm for 120 min, they formed large aggregates in the dark but remained single cells under blue light (Fig. 2a). As a measure of the cell−cell adhesions, we determined the average cluster area in the samples, where clusters were defined as an object with an area >5000 μm$^2$. Indeed, in the dark and under blue light, the cell aggregates had an average projected area of $2.3 \times 10^4$ and $9.2 \times 10^3$ μm$^2$, respectively. In addition, in this cell culture system, cell aggregation

required the $Ca^{2+}$ site to be occupied, as larger aggregates failed to form in the presence of $Tb^{3+}$ even in the dark (Supplementary Fig. 3c).

This raises the question about the range of $Ca^{2+}$ concentrations that opto-E-cad operates in, as the insertion of the AsLOV2 domain might also alter $Ca^{2+}$ binding affinity in the dark. When we repeated the cell aggregation assay at different calcium concentrations up to 2 mM, the opto-E-cad-MDA cells failed to aggregate in the absence of $Ca^{2+}$, irrespective of the illumination status (Fig. 2c, Supplementary Fig. 3d). At $Ca^{2+}$ concentrations above 0.5 mM, the cells aggregated to the maximal extent in the dark, but remained as single cells under blue light. This range of extracellular $Ca^{2+}$ concentration (0.5–2 mM) is similar to what is found in most common cell culture media and therefore would allow the opto-E-cad to be implemented in many cell culture systems. To demonstrate that E-cadherin or the MDA-MB-231 cells are not influenced by blue light illumination, we created a monoclonal cell line from MDA-MB-231 cells by transfecting only the E-cadherin plasmid without the inserted LOV2 domain (E-cad-MDA). Although these E-cad-MDA cells formed clusters of sizes similar to those of the opto-E-cad cells in the dark, clusters of the same size were also formed under blue light (Supplementary Fig. 3e).

Opto-E-cad provides many possibilities for modulating cell–cell adhesions. One way to control the degree of adhesion is through different intensities of blue light. We found that the aggregate sizes can be turned from large aggregates in the dark to single cells by varying the blue light illumination from 0 to 272 μW/ $cm^2$ (Supplementary Fig. 5a). Considering the wavelength specificity, we observed that the opto-E-cad adhesions specifically were turned off only under blue light, whereas samples incubated under red light aggregated to similar levels seen in the dark (Supplementary Fig. 5b). The integration of the cofactor is essential for the function of the LOV2 domain in the opto-E-cad and we found that the addition of the cofactor to the culture media was essential for the function (Supplementary Fig. 5c). Another factor that alters cell adhesions is the expression level of opto-E-cad on the cell surface. Different monoclonal opto-E-cad-MDA cell lines with various expression levels ($7 \times 10^3$ to $7 \times 10^5$ opto-E-cad molecules per cell as quantified by flow cytometry) formed larger cell clusters with increasing expression levels in the dark, and the differences in aggregate size in the dark and under blue light became significant for clones with an expression level above $1 \times 10^4$ opto-E-cad molecules per cell (Supplementary Fig. 5d, e). The opto-E-cad-MDA clone used for all other experiments had approximately $7.4 \times 10^5$ opto-E-cads per cell on the plasma membrane.

Opto-E-cad can also be employed in other cell types to make their cell–cell adhesion photoswitchable. We generated monoclonal stable cell lines transfected with opto-E-cad from HeLa cells, A431D cells that have a specific E-cadherin knockout and L-929 fibroblast cells. In all cell lines with opto-E-cad expression, we observed significantly larger cell clusters in the dark than under blue light or in those formed by the parent cell line (Supplementary Fig. 6a, b). All of the here tested cell lines have low native E-cadherin expression and the opto-E-cadherin is best used in cells with a low E-cadherin background or specific E-cadherin knockouts.

## Bidirectional switching of cell–cell adhesions with opto-E-cad

Opto-E-cad provides high temporal and reversible control over cell–cell adhesions (Fig. 3a). In suspension cultures, the opto-E-cad-MDA cells required about 60 min in the dark to form larger multicellular aggregates (a 1.3-fold difference in cluster area compared with that for cells kept under blue light), which increased to a 3-fold difference in cluster size after 180 min (Fig. 3b). The aggregates in the dark increased in size over a couple of hours and reached a plateau. Their size was similar to that of aggregates formed by MCF-7 cells, which highly express E-cadherin and showed pronounced cell aggregation as well. In contrast, the opto-E-cad-MDA cells under blue light

and the parent MDA-MB-231 cells did not cluster significantly even after 180 min.

A hallmark of MET and EMT is the temporal up and down regulation of E-cadherin-based cell–cell adhesions[8]. To demonstrate that opto-E-cad can be used to induce cell–cell adhesions and reverse them as desired, we first kept opto-E-cad-MDA cells under blue light for 60 min and then placed them in the dark. Here, we observed an increase in cell aggregation only for cells after turning the illumination off and cells kept under illumination remained as single cells (Fig. 3c). Conversely, for the cells first kept in dark for 60 min and then exposed to blue light, the cluster area decreased rapidly within 30 min to the level observed in a positive control where the cell–cell adhesions were disrupted with ethylenediaminetetraacetic acid (EDTA) (Fig. 3d). In contrast, cells preincubated in the dark for 120 min did not show the same dissociation of the aggregates when placed under blue light within 180 min (Supplementary Fig. 7a, b). This difference could potentially be due to secondary interactions formed between the cells, stabilization of the adhesions over time or shear forces (30 rpm) in the suspension culture not being strong enough to separate cells once they have clumped together. Yet, in experiments detailed below, we observed reversion of the opto-E-cad adhesions upon blue light illumination even after 2 days of prior culturing in the dark. The opto-E-cad based cell–cell adhesions could also be switched on and off repeatedly as shown in three dark/blue light cycles with one-hour intervals (Fig. 3e). In these experiments there was no sign of fatigue in the photoswitching.

## Light-controlled intracellular E-cadherin activity

Next, we analyzed whether opto-E-cad-mediated intercellular adhesions result in actin cytoskeleton reorganization and recruit catenins to the intracellular tail (Fig. 4a)[35]. Opto-E-cad-MDA cells kept overnight in 2D cell culture had different F-actin arrangements, depending on the illumination (Fig. 4b). In the dark, the F-actin was cortical and arranged in for epithelial cells typical cobblestone pattern, as also observed in the MCF-7 cells (positive control). In contrast, in opto-E-cad-MDA cells kept under blue light, F-actin stress fibres were observed, similar to those in the parent MDA-MB-231 cells (negative control). Moreover, the average cell spreading area of the opto-E-cad-MDA cells growing in clusters in the dark was significantly smaller than single cells in the culture (Fig. 4c). In comparison, MDA-MB-231 cells had a similar spreading area to single opto-Ecad-MDA cells and E-cad-MDA cells (MDA-MB-231 cells transfected with E-cadherin (E-cad-MDA) used as a positive control) had a comparable spreading area to opto-E-cad-MDA cells growing in clusters.

In parallel, the localization of p120, which is one of the catenins that binds to the cytoplasmic tail of E-cadherin upon forming cell–cell adhesions, showed major differences under different culture conditions. In the dark, p120 localized at the cell–cell adhesion sites and was also visible in a cobblestone pattern (Fig. 4b). Yet upon blue illumination overnight, the p120 signal was lower in the cells and the staining was sparser. Likewise, in 3D cellular aggregates of opto-E-cad-MDA cells formed in the dark, the p120 staining was also clearly visible at the cell–cell junctions (Fig. 4d). Complementarily, we observed the expression of opto-E-cad in the opto-E-cad-MDA cells both in the dark and under blue light overnight in western blots (Fig. 4e). Further, in agreement with the immunostainings, we observed a stronger band for p120 for opto-E-cad-MDA cells in the dark than under blue light. In fact, the opto-E-cad-MDA cells in the dark had a similar p120 expression to that of E-cad-MDA cells; and those kept under blue light lacked the expression like the parent MDA-MB-231 cells (negative control). p120 catenin has been shown to have an important function in stabilizing E-cadherins at adhesion sites and is degraded when it is not associated with the adhesions[36]. The light dependent catenin p120 levels and localization are directly associated with the light-regulated cell–cell adhesions, and they support the molecular picture that the

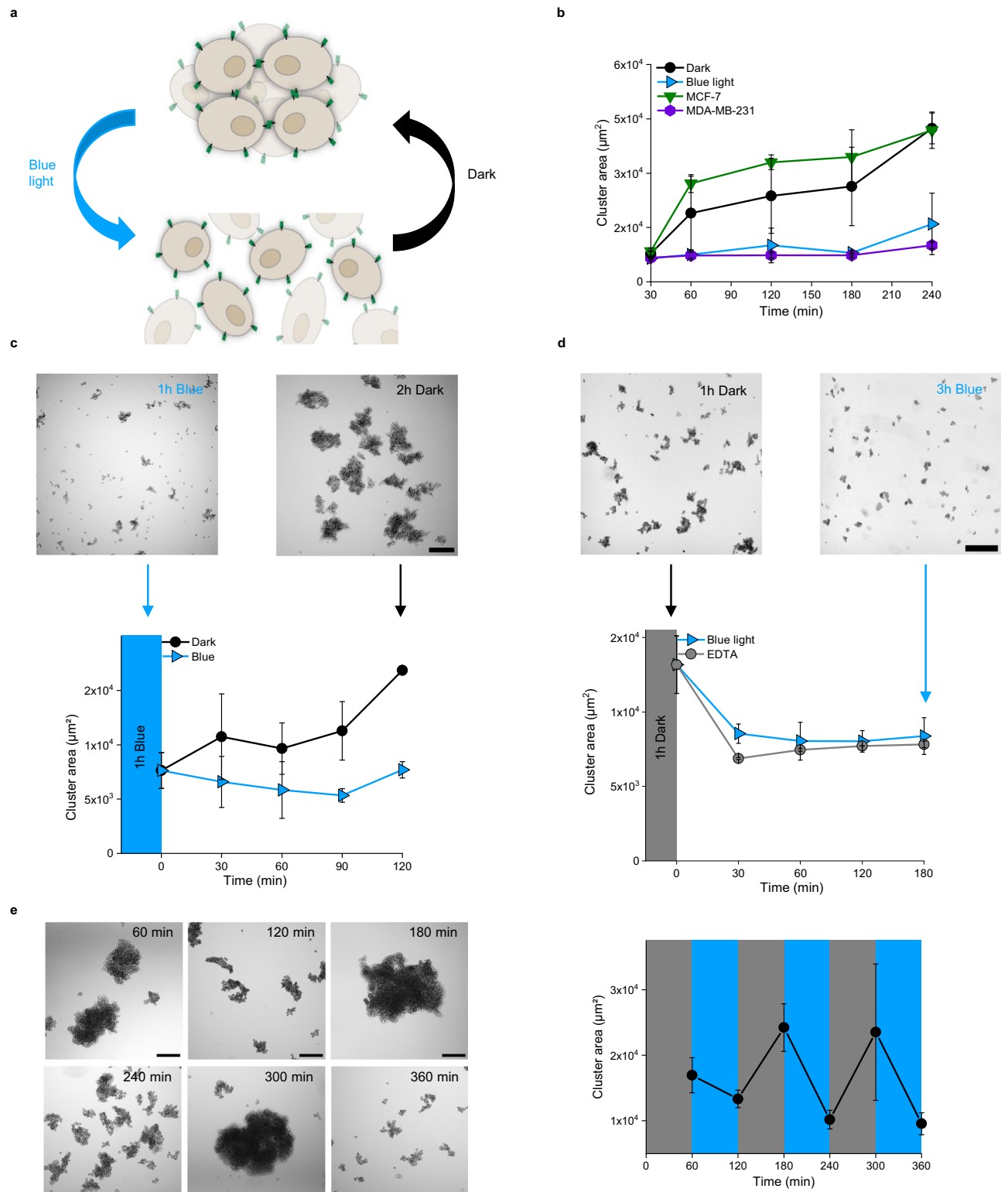

**Fig. 3 | Temporal and bidirectional control over cell–cell adhesions. a** Opto-E-cad allows to turn adhesions on upon placing cells in the dark and off upon blue light illumination. **b** Quantification of average cell cluster area over time in suspension culture in the dark and under blue light (*n* = 6). **c** Bright field images of opto-E-cad-MDA cells in suspension culture kept for 60 min under blue light and subsequently in the dark or blue light and formation kinetics of the cell–cell adhesions (*n* = 4). **d** Bright field images of opto-E-cad-MDA cells in suspension culture kept for 60 min in the dark and subsequently under blue light and reversion kinetics of cell–cell adhesions under blue light (*n* = 8). **e** Bright field images of opto-E-cad-MDA cells in suspension culture under repeated 60 min dark/blue light cycles and their clustering dynamics (*n* = 3). Data are represented as mean values ± SD. Scale bars are 200 µm. Source data are provided as a Source Data file.

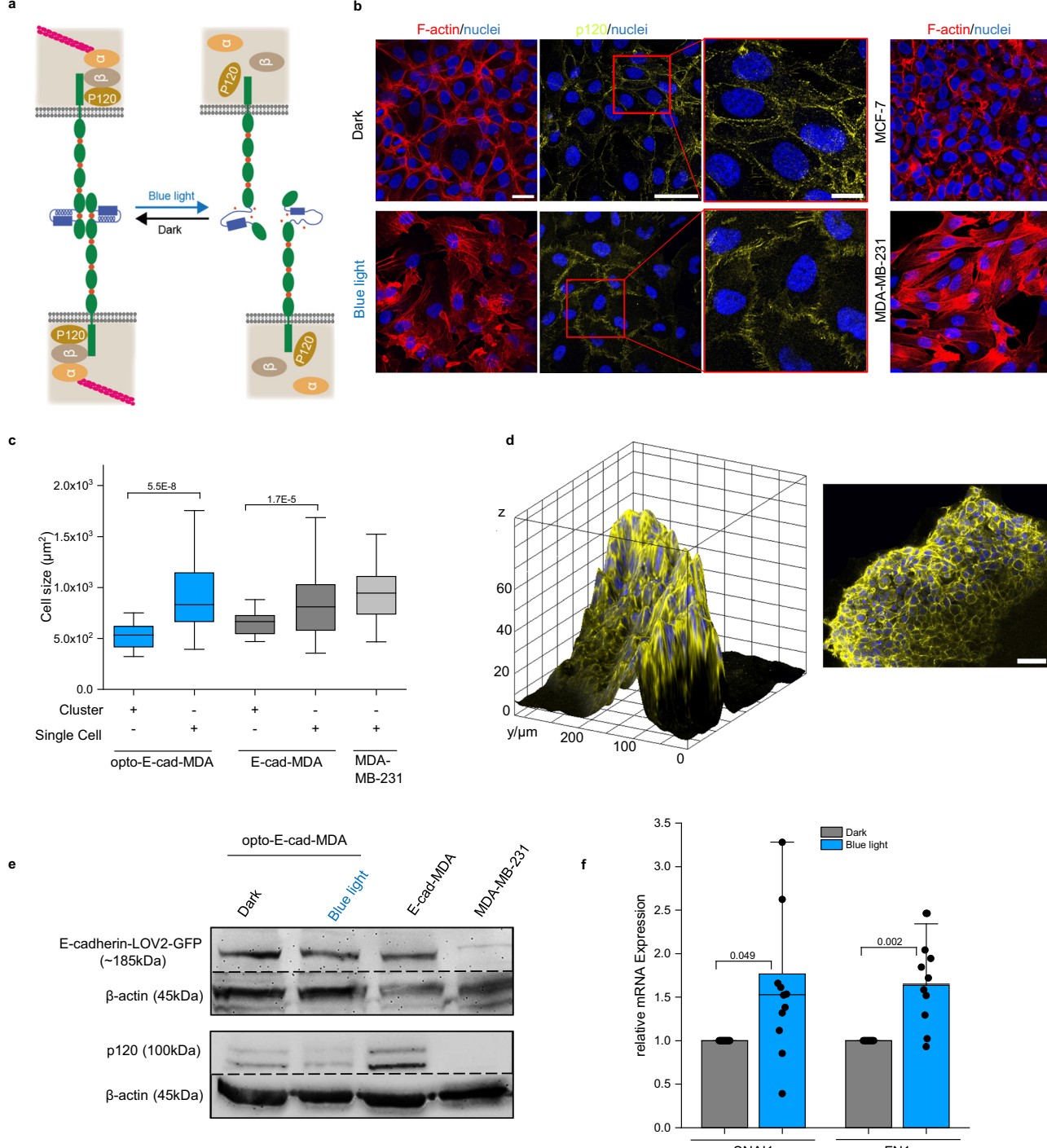

**Fig. 4 | Light-controlled intracellular event with opto-E-cad. a** Schematic representation of opto-E-cad-dependent cell signalling. In the dark, the opto-E-cads on adjacent cells homophilically bind to each other, and the catenins (p120, α- and β) are recruited to the intracellular tail and link it to F-actin. Under blue light, opto-E-cads do not form cell–cell adhesions, no catenins are recruited and there is no link to the actin cytoskeleton. **b** Confocal fluorescence microscopy images of opto-E-cad-MDA cells in the dark or under blue light (MDA-MB-231 and MCF-7 cells in 2D cultures). F-actin is shown in red, nuclei are shown in blue and p120 is shown in yellow. Scale bars are 50 μm and 15 μm for the inset. **c** Cell spreading area of opto-E-cad cells forming cell–cell adhesions in cluster compared to single cells. $n$ = 35, 46, 14, 46, 46. Boxplots show median, 25th, and 75th percentile. The lower and upper boundaries of whiskers indicate the minima and maxima, respectively. All comparisons are performed using Fisher's One Way ANOVA test and $p < 0.05$ was treated as the significance threshold. **d** 3D reconstruction and maximal intensity projection for p120 (yellow) and nuclei (blue) of opto-E-cad-MDA cell clusters after 120 min in the dark. Scale bar is 50 μm ($n$ = 3). **e** Western blot of E-cadherin, p120 and β-actin (loading control) for opto-E-cad-MDA in the dark or under blue light and E-cad-MDA and MDA-MB-231 cells as positive and negative control, respectively ($n$ = 4). **f** RT-PCR results for opto-E-cad-MDA in the dark and under blue light for EMT markers. (two-tailed t-test, $n$ = 12 and 10). Data are represented as individual values ± SD. Source data are provided as a Source Data file.

mechanosensitive binding of p120 to the intracellular tail of opto-E-cad and is upregulated in the dark, while under blue light the interaction with the intracellular tail of opto-E-cad is lost and p120 is degraded.

Finally, we investigated if photoregulation of the opto-E-cad alters E-cadherin associated cellular signalling. For this purpose, we measured the upregulation of EMT markers in opto-E-cad-MDA cells after overnight incubation under blue light illumination using RT-PCR (Fig. 4f). The mRNA levels for both the EMT master regulator Snai1 and the extracellular matrix protein fibronectin-1 increased under blue light illumination compared to cells kept in the dark. Overall, these results show that the opto-E-cad allows switching to a more mesenchymal phenotype under blue light illumination compared to darkness, changing the connection to the actin cytoskeleton, the intracellular interactions with catenins and the gene expression profile.

## Photoregulating collective cell migration and invasion with opto-E-cad

Cell–cell adhesions are essential for collective cell migration, as they transmit both mechanical and biochemical signals that allow for coordinated movement[37]. When cells lose contact with their neighbours, they can move away from their tissue of origin and migrate as individuals, as seen in metastasis[10]. Here, we investigated whether opto-E-cad can be used to switch between collective and single cell

migration and to change invasive behaviour of cells with light. For this purpose, we used a wound healing assay, where a confluent layer of opto-E-cad-MDA cells grown in the dark was wounded and their migration into the open space analyzed in the dark and under blue light. We found that in the dark, the opto-E-cad-MDA cells migrated collectively, forming a stable and coordinated front (Fig. 5a, Supplementary Movie 1). In contrast, under blue light illumination, single cells migrated individually and without a preferred direction into the wound and the wound edge fringed over time (Supplementary Movie 2). This difference in migration modalities was also reflected in the migration rates of the wound edge, which were 3.24 nm/s and 1.59 nm/s in the dark and under blue light, respectively (Fig. 5b). Two parameters that measure collective migration are the correlation length between the cells and the migration angle of the cells. The correlation length was higher in the dark than under blue light, showing better coordination between cells in the dark (Fig. 5c). In comparison, the parent MDA-MB-231 cells (negative control) showed no light dependence for correlation length and had a correlation length comparable to that of the opto-E-cad-MDA cells under blue light. Similarly, in the dark, the migration angles of the opto-E-cad-MDA cells were between 0 and 90 degrees, showing movement directly into the wound, whereas under blue light, a smaller fraction of cells had a migration angle between 0 and 90 degrees, showing less

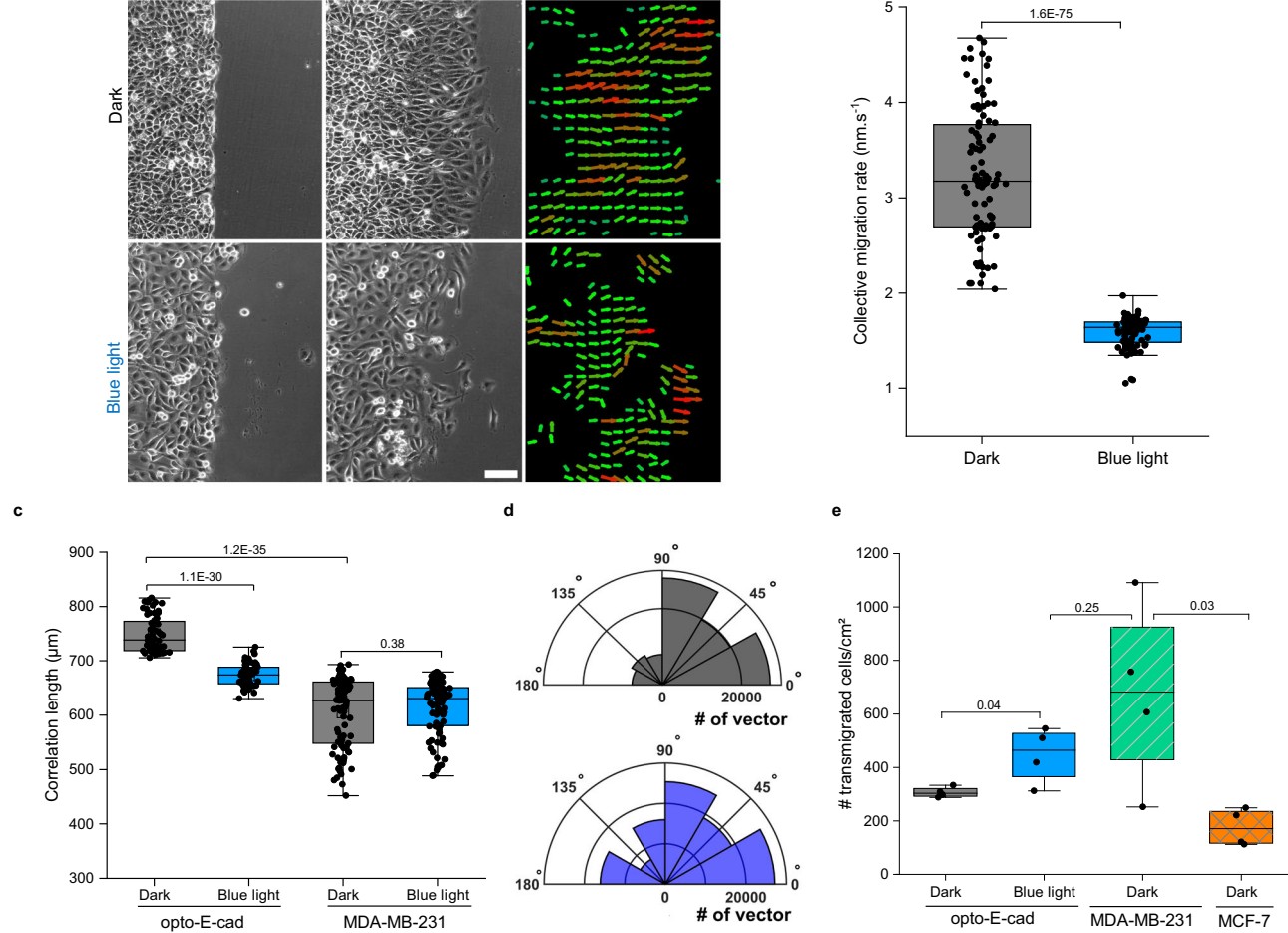

**Fig. 5 | Optogenetic control of collective cell migration and cell invasion.**
**a** Bright field and vector field images of opto-E-cad-MDA cells in a wound healing assay in dark and under blue light for 16 h. Scale bar is 100 µm. 6 individual experiments were conducted. **b** Migration rate of the cell front for opto-E-cad-MDA cells (**a**) ($n = 33$). All comparisons are performed using Fisher's One Way ANOVA test and $p < 0.05$ was treated as the significance threshold. **c** Velocity correlation lengths

of opto-E-cad-MDA and MDA-MB-231 cells in the dark and under blue light, from left to right $n = 64, 64, 93, 93$ (two-tailed *t*-test). **d** Migration angle of velocity vectors of opto-E-cad-MDA in dark (upper liner) and under blue light (lower line). **e** Transwell invasion assay (two-tailed *t*-test, $n = 4$). Boxplots (**b**, **e**) show median, 25th, and 75th percentile. The lower and upper boundaries of whiskers indicate the minima and maxima, respectively. Source data are provided as a Source Data file.

coordination in the migration (Fig. 5d). In addition, the opto-E-cad provided high spatiotemporal control over cell–cell adhesion and allowed local induction of different cellular migration behaviours. When we partially illuminated an area in the wound, individual cells migrated randomly out of the monolayer and the wound front fragmented in these parts, whereas the cells in the dark retained a stable and coordinated front and moved forward together (Supplementary Fig. 8). Overall, the opto-E-cad-MDA cells adhered strongly and migrated collectively in the dark, whereas they migrated individually in random directions under blue light.

Cell–cell adhesions also altered the invasiveness of the cell, which is of relevance during cancer metastasis. To assess the invasive properties of opto-E-cad-MDA cells, we used the Transwell Cell Migration Assay, in which cells migrate from one well to another separated by a semipermeable membrane following a chemo-attractant. We observed that opto-E-cad-MDA cells were less invasive in the dark than under blue light (Fig. 5e, Supplementary Fig. 9). In fact, the invasiveness of the opto-E-cad-MDA cells was similar to that of MDA-MB-231 cells, which are a highly invasive breast cancer line. The stronger cell–cell adhesion of the opto-E-cad-MDA cells in the dark than under blue light makes it more difficult for the cells to go through pores that allow only individual cells to penetrate. Blue light induced single cell migration and thus promoted individual cell migration through the fixed-diameter pores of the transwell filters.

### Opto-E-cad controls cell–cell adhesions in 3D cell culture

Next, we investigated whether opto-E-cad was functional and could dynamically photoregulate cell behaviour in 3D cell culture systems. In particular, we used spheroids in which compacting depends on cell–cell adhesions (Fig. 6a). When spheroids were formed in the dark, opto-E-cad-MDA cells (stained with CellTracker Green, shown in green) formed compact spheroids, whereas under blue light, the cells remained as loose branched aggregates. This difference was also reflected in the volume of the spheroids. In the dark, opto-E-cad-MDA cells compacted to a smaller volume ($0.024\ mm^3$) than they did under blue light ($0.037\ mm^3$) (Fig. 6b). Similarly, also spheroids formed form opto-E-cad-A431D cells were also smaller in volume when kept in the dark than under blue light (Supplementary Fig. 10). The compacting of these spheroids correlated with the number of active E-cadherins, as also reflected in the large volume of spheroids formed from MDA-MB-231 cells ($0.098\ mm^3$, negative control) and the very compact spheroids formed from MCF-7 cells ($0.008\ mm^3$, positive control). The fact that the spheroid volume remained the same in the dark and under blue light for MDA-MB-231 is further proof of the light-dependent compacting in opto-E-cad-MDA cells being due to their photoswitchable cell–cell adhesions and not an artefact of light illumination. The higher compactness of opto-E-cad-MDA spheroids under blue light than MDA-MB-231 spheroids indicate that even in the dark that there is some residual attraction between opto-E-cad-MDA cells. As no switching is complete some background activity is always expected and this background may contribute differently in various assays depending on the sensitivity. For example, in the 3D clustering assay the cells are mildly agitated at 30 rpm, which may already rupture weak interactions. In contrast, for spheroid formation, the cells are centrifuged to the bottom of the U-well and kept still in the incubator in a non-adhesive environment. In the absence of other forces (e.g. cell-matrix, shear forces) even weak cell–cell interaction lead to some spheroid compacting.

### Controlling cell invasion in 3D cultures

An important strength of the opto-E-cads is that cell–cell adhesions can be altered during the experiment simply by altering illumination. We took advantage of this property to investigate how cells invade a 3D extracellular matrix, thereby recapitulating the dissemination of E-cadherin-bearing carcinoma cells from their primary tumour node and

metastasizing to other parts of the body. Towards this end, in an in vitro system, compact opto-E-cad-MDA spheroids were first formed in the dark as described earlier and, after 1 day, were embedded in type I collagen gels. If at this point the samples were kept in the dark, only a few cells invaded the collagen matrix, and the embedded spheroids remained compact even after 2 days (Fig. 6c). However, if the samples were illuminated with blue light, the cells were highly invasive, and many detached from the spheroid core and moved into the collagen gel as single cells. For comparison, MDA-MD-231 cells, which are known to be very invasive and unable to form compact spheroids, also moved and spread completely into the collagen gel, like the opto-E-cad-MDA cells did under blue light. In contrast, MCF-7 cells remained as compact spheroids and only a few cells invaded the gel, similar to the opto-E-cad-MDA cells in the dark. In this experimental set-up, the opto-E-cad mediated adhesions were reversed after 2 days when the spheroids were placed in the collagen gels. In this case, the strong cell-matrix adhesions may support the cells to overcome residual cell–cell adhesions and invade into the matrix as single cells.

To assess the invasion behaviour, we defined two parameters: the number of cells invading the gel and the area of the remaining spheroid core. These two parameters are interconnected; a high number of invading cells results in a smaller core area and a low number of invading cells is associated with a larger core area. For opto-E-cad-MDA cells, after 2 days, the number of invading cells was about 2-fold higher under blue light than it was in the dark (Fig. 6d). At the same time, the core area of the spheroid was 2-fold larger in the dark than under blue light (Fig. 6e). In contrast, the controls with MDA-MB-231 and MCF-7 cells showed no significant difference in the dark and under blue light (Fig. 6d, e, Supplementary Fig. 11a). Overall, these results show that opto-E-cad is able to mimic the increased cell invasion that takes place during cancer cell spread from the primary tumour, where cells lose the adhesions to their neighbours.

### Opto-E-cad controls cell–cell adhesions in vivo

To translate the findings from the in vitro invasion assay to an in vivo tumour dissemination model, we assessed opto-E-cad-MDA cell invasion by using the chick chorioallantoic membrane (CAM) assay. The CAM assay allows implantation of cells into the extraembryonic membrane of a developing chick embryo and is an accessible animal model for the investigation of tumour progression in a physiological environment[38]. Here, we inoculated opto-E-cad-MDA spheroids after 2 days in the dark onto the CAM to observe the light-dependent cell invasion. After 1 day on the CAM in the dark, the opto-E-cad-MDA cells (shown in green) remained in the spheroids and not many single cells were observed invading the CAM (Fig. 6f, Supplementary Fig. 11b). Yet, if kept under blue light, already after 1 day, the opto-E-cad-MDA cells no longer formed spheroids, and the single cells invaded the CAM. In confocal fluorescence images of samples where all nuclei were stained (shown in blue), the spheroids only loosely attached to the chorionic ectodermal surface of the CAM (filled arrow heads) in the dark with no cells detaching from spheroids and infiltrating the CAM (Fig. 6g). In contrast under blue light, the cells invaded the entire CAM all the way down to its allantoic endoderm side (open arrow heads) and opto-E-cad-MDA cells intermixed with the surrounding cells. In comparison, the MCF-7 spheroids (positive control) remained intact and the MDA-MB-231 spheroids (negative control) were highly invasive, independent of being in the dark or under blue light illumination (Supplementary Fig. 11c). These results showed that opto-E-cad-MDA cells behave like MCF-7 cells in the dark and like MDA-MB-231 under blue light and allowed us to dynamically turn E-cadherin-mediated cell–cell adhesions on and off in different cell culture systems and in vivo.

## Discussion

Here, we presented a powerful optogenetic tool for the reversible and spatiotemporally controlled regulation of E-cadherin-mediated

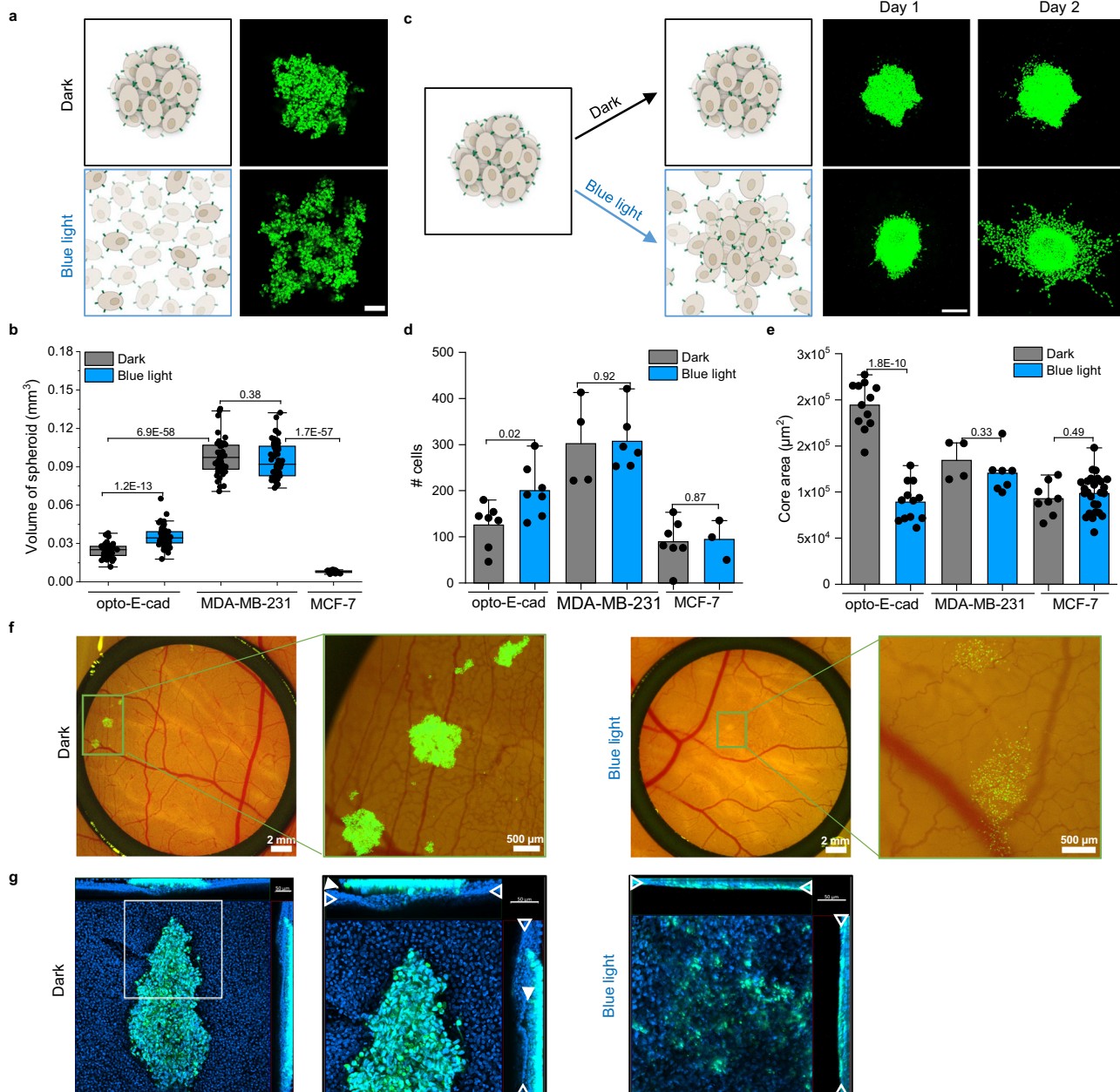

**Fig. 6 | Light-dependent spheroid formation and light-controlled cell invasion in 3D matrices. a** Schematic representation (left panel) and confocal fluorescence images (right panel) of spheroids formed from opto-E-cad-MDA cells in the dark and under blue light. Cells stained with CellTracker Green shown in green. Scale bar is 100 μm. 4 individual experiments were conducted. **b** Spheroid volume after 24 h (two-tailed *t*-test, *n* = 54). Boxplots show median, 25th, and 75th percentile. The lower and upper boundaries of whiskers indicate the minima and maxima, respectively. **c** Schematic representation (left panel) and confocal fluorescence images (right panel) of spheroids (preformed in the dark) embedded in a type I collagen hydrogel in the dark and under blue light. In the absence of cell–cell adhesions under blue light, the cells invade the matrix. Cells were stained with CellTracker Green shown in green. Scale bar is 200 μm. 6 individual experiments were performed. **d** Number of cells (two-tailed t-test, from left to right *n* = 7, 7, 4, 6,

7, 3) and **e** the area of the spheroid core for cells from spheroids invading a type I collagen hydrogel in the dark and under blue light after 48 h (two-tailed t-test, from left to right *n* = 12, 12, 4, 7, 8, 26). Data are represented as mean values ± SD. **f** Fluorescence and bright field images of opto-E-cad-MDA spheroids (preformed in the dark, cells shown in green) 24 h after inoculation on a CAM. 3 individual experiments were conducted. **g** Confocal fluorescence images showing opto-E-cad-MDA cells in green and nuclei in blue after 24 h on the CAM (*n* =1). In the dark, cell–cell contacts prevented the disintegration of the spheroids, which loosely attached to the chorionic ectodermal surface of the CAM (arrows), with no cells detaching from spheroids and infiltrating the CAM. Under blue light, the cells invaded the entire CAM all the way down to its allantoic endoderm side (arrow heads). Scale bars are 50 μm. Source data are provided as a Source Data file.

adhesions at the molecular, cellular and behavioural level. The opto-E-cad with a photoswitchable AsLOV2 domain has several advantages over existing systems that involve light-responsive small molecules or optogenetic systems that rely on light-dependent dimerization. The fact that the opto-E-cad is a one-component optogenetic tool avoids all the problems that can arise from two or more component systems

where the expression levels of different constructs have to be matched. The LOV2 domain can undergo repeated conformational changes over many blue light/dark cycles, which allows one to dynamically turn the opto-E-cad adhesions off and on again without using cell-toxic UV light. Moreover, the cofactor of the LOV2 photoswitchable domain does not require synthetic small molecules that could degrade,

undergo a change in concentration or have off-target effects. The entire system is genetically coded and therefore the adhesion molecules are constantly maintained over a long period. These properties allowed us to study and photoregulate cell–cell adhesions in various 2D and 3D cell culture systems over many days. The opto-E-cad construct is highly transferable into different cell types and could even lead to applications in optically transparent model organisms amenable to genetic manipulation in the future.

In designing opto-E-cad, we targeted structurally and functionally critical $Ca^{2+}$ binding sites and altered their $Ca^{2+}$ binding ability in a light-dependent way. This design essentially provides an ON/OFF dial for cell–cell adhesions. In the ON state in the dark, the C-terminal Jα-helix of LOV2 remains folded and does not disturb the $Ca^{2+}$ binding site between EC1 and EC2, such that adjacent cells are able to form stable homophilic cell adhesions. In the OFF state under blue light, the partial unfolding of the LOV2 domain results in $Ca^{2+}$ binding amino acids becoming part of an unstructured domain, as well as a loss in cell–cell adhesions. Conventionally, $Ca^{2+}$ chelators such as EDTA are added to disrupt calcium-dependent cell adhesions, including E-cadherins, but at the same time, they affect other cadherins, integrins and selectins with no specificity. Opto-E-cad provides the desired molecular specificity in targeting the $Ca^{2+}$ binding site and operates at $Ca^{2+}$ concentrations in the low millimolar range commonly found in the extracellular environment and in cell culture media. Considering that many cadherins have high structural homology, we expect the design principles of the opto-E-cad to be transferable to other types of relevant cadherins that are equally dynamically regulated throughout physiological and pathological processes.

Directly regulating extracellular binding between E-cadherins on adjacent cells, opto-E-cad enables control over E-cadherin-based adhesions and all downstream processes. When we switched the cells from dark to blue light illumination, we observed changes at the cellular level in the reorganization of the actin cytoskeleton, the localization of the E-cadherin-associated p120 catenin and the expression of mesenchymal markers as well as at the behavioural level in 2D cell culture in cell clustering and a switch from collective to individual cell migration. Further, in 3D cell culture and in vivo, we observed changes in spheroid compacting, invasion capacity and migration into collagen hydrogels and in vivo. All these changes in the cell characteristics mimic the loss of E-cadherin typically observed during EMT and show how we are able to trigger EMT not through the addition of soluble chemicals, but by using light. Yet, with light as a stimulus, we were able to spatially and temporally control this transition in cell characteristics and even reverse it by simply turning off the light, as was the case in the mesenchymal-epithelial transition. Therefore, opto-E-cad is an excellent tool to analyze changes in cell signalling, migration and invasion related to E-cadherin dynamics.

E-cadherin dynamics are essential in a multitude of biological processes, such as embryonic development, wound healing and cancer development[39], and the opto-E-cad could provide a deeper mechanistic insight into these processes. First, the homotypic E-cadherin adhesions in epithelial tissues are important for initiating cell recognition, organizing cells into tissues, maintaining tissue integrity and exerting cell sheet forces during development[40]. Yet, the dynamics of cell–cell adhesions still allow these cells to move with respect to each other, leading to tissue plasticity[41] and regulating cell proliferation, survival, invasion and migration[42]. During embryo morphogenesis, controlled and coordinated cell movements are even more pronounced, and one of the major driving forces that lead to germ layer arrangement is cell–cell adhesions that are spatiotemporally regulated[43]. The herein demonstrated ability of opto-E-cad-expressing cells to change with light illumination cell clustering in 2D suspension and spheroid cultures, as well as their downstream signalling and migration behaviour, reveals that opto-E-cad allows the external manipulation of these events.

Finally, the loss, reduction or dysfunction of E-cadherins is consistently observed in most progressive, aggressive and undifferentiated carcinomas of the mammary gland and other epithelial tissues[44]. The microenvironment of such tumours is characterized by mechanical properties distinct from those of healthy tissues, including a stiffer extracellular matrix that is mainly composed of collagen[38]. The in vitro and in vivo invasion assays demonstrated how we can use opto-E-cad-expressing cells to show the impact that changes in cell–cell adhesions have on cell migration and invasiveness. As a relevant model for this process, we embedded into collagen gels spheroids of opto-E-cad-MDA cells prepared in the dark and investigated their invasion in the 3D matrix in the dark and under blue light. Despite the parent MDA-MB-231 cells being a highly invasive breast cancer cell line, the opto-E-cad-MDA cells remained as compact spheroids in the dark and did not invade the collagen gel, because of intercellular adhesions. Only when the adhesions between the cells were turned off through the illumination with blue light did the opto-E-cad-MDA cells became invasive and migrated as individual cells into the collagen matrix. These observations were also mirrored in the CAM assay, which is well suited for investigating tumour invasion and metastasis in vivo in a highly vascularized and physiological extracellular matrix that includes fibronectin, laminins and type I collagen[45].

Overall, opto-E-cad is a powerful optogenetic tool capable of controlling cell–cell adhesions at the cellular, molecular and behavioural level and functions as a light-dependent binary switch in 2D and 3D cultures. Opto-E-cad opens opportunities to control the spatiotemporal dynamics of E-cadherins, which play a central role in numerous biological processes.

## Methods

### Opto-E-cad plasmid preparation

E-cadherin-GFP plasmid (a gift from Jennifer Stow, Addgene plasmid # 28009)[46] codes for the full-length human E-cadherin with a C-terminal enhanced GFP fusion in a pcDNA3.1 vector with a geneticin selectable marker. The AsLOV2 domain was amplified from the pET21b-LOV-ipaA plasmid (a gift from Brian Kuhlman, Addgene plasmid # 40236)[47] with modifications at the C-terminus in the primers and introduced between T133 and D134 of E-cadherin-GFP by using Gibson assembly (see primers below), resulting in the opto-E-cad plasmid (see sequence below).

### Cell culture and generation of monoclonal opto-E-cad cell lines

The breast cancer cell lines MDA-MB-231 and MCF-7, HeLa and the fibroblast cell line L929 were obtained from the American Type Culture Collection. The A431D cells were kindly provided by Dr. Ada Cavalcanti-Adam (Department of Cell Research, Max Planck Institute for Medical Research, Heidelberg, Germany). The cells were cultured in Dulbecco's Modified Eagle Medium (DMEM, PAN Biotech, #04-03591) without phenol red, supplemented with 10% foetal bovine serum (FBS, PAN Biotech, P30-3031), 1% penicillin–streptomycin (10,000 U/mL) (Gibco, #15140122) and 12.5 mM HEPES (Sigma-Aldrich, H4034-500G) at 37 °C and 5% $CO_2$.

To generate stable cell lines, cells were transfected with the opto-E-cad plasmid by using Lipofectamine 3000 (Thermo Fisher Scientific, L300001), following the manufacturer's protocol for a 24-well plate. The transfected cells were selected with 1800 µg/mL geneticin (Geneticin G418, Roche, 4727878001) and maintained in the presence of geneticin for all further experiments. After the cells were cultured for 1 week with geneticin selection, they were FACS sorted (BD FACS Aria cell sorter) into a 96-well plate with one cell per well based on the fluorescence of the tagged GFP. Following weeks of expanding monoclonal cultures, GFP fluorescence was assessed by using confocal microscopy and FACS. The monoclonal cell lines of opto-E-cad with the highest protein expression were used in further experiments.

## Quantification of protein expression on the cell plasma membrane

Opto-E-cad-MDA and MDA-MB-231 cells were cultured to 80% confluence and washed twice with phosphate-buffered saline (PBS) (Gibco, 18912014). Afterwards, the cells were detached for 10 min at room temperature with Accutase (StemPro Accutase Cell Dissociation Reagent, Gibco, A1110501) diluted 1:4 in Hanks' Balanced Salt Solution (HBSS, no calcium, no magnesium, no phenol red, Gibco, 2549069) to minimize E-cadherin degradation. Subsequently, the cells were collected by centrifugation, suspended in PBS and counted with an automated cell counter (Bio-Rad TC20TM). From each cell type, $1 \times 10^6$ cells in 200 μL PBS were incubated with the primary antibody mouse anti-E-cadherin antibody (cell signalling # 14472, dilution 1:1000) and incubated at 4 °C while being gently mixed on the shaker for 1 h. Following this, the cells were washed three times by adding 800 μL of cold PBS to them and recovering the cells with centrifugation ($600 \times g$, 4 °C for 5 min). The cells were resuspended in 200 μL PBS and incubated with Alexa Fluor 488 goat anti-mouse IgG (Invitrogen, # A11029, dilution 1:1000) at 4 °C on the shaker for 1 h. The cells were washed three times as described earlier, resuspended in 400 μL of PBS and analyzed by using flow cytometry (BD FACS Celesta). The Quantum Alexa Fluor 488 MESF kit (Bang Laboratories, Inc., 488A) was used as a standard for quantification following the manufacturer's protocol. The median fluorescence peak from each cell type was measured with FlowJo and converted into MESF (molecules of equivalent soluble fluorochrome) on the basis of the calibration curve generated with Quick Cal v.2.4 software from Bang Laboratories. The signal from MDA-MB-231 cells (negative control) was subtracted as background from the signal of the opto-E-cad-MDA cells.

## Light sources and toxicity

In light experiments for cell aggregation and the migration assay, blue or red light LED panels (225 LEDs, 463 nm 272 μW/cm$^2$ or 620 nm 1440 μW/cm$^2$) were used with one or more neutral-density filters (50% reduction). A CLF flora LED module with a controller (CLF Plant Climatics GmbH) was used (463 nm, 20.4 μW/cm$^2$) in the 2D cell clustering assay, different invasion assays, spheroid cultures, the CAM assay and samples prepared for immunofluorescence and western blotting. All the dark samples were continuously wrapped in aluminium foil.

To test the toxicity of the blue light in different experimental setups, we seeded $5 \times 10^4$ cells/well in 100 μL medium into 96-well microplates (Greiner bio-one, CELLSTAR; # 655 180) and exposed them to different light intensities (0.2–240 μW/cm$^2$) for the maximum duration and under the same conditions as the corresponding experiment. Afterwards, 10 μL of 0.5 mg/mL MTT reagent was added to each well, the cells were incubated for 4 h at 37 °C and 5% CO$_2$ and then 100 μL of MTT solvent (4 mM HCl and 0.1% NP40 (nonyl phenoxypolyethoxylethanol) in isopropanol) was added to each well. The microplate was then wrapped in aluminium foil and incubated on an orbital shaker for 15 min. Subsequently, the absorbance of the formazan product at 590 nm was measured.

## Light-responsive cell–cell adhesions in 2D culture

Prior to the experiment the cells were seeded into T25 flasks at 20–30% confluency and cultivated for 3 days. Cells were washed with PBS and detached with 0.5 mL Accutase diluted 1:4 in HBSS at room temperature for 10 min, and then harvested by centrifugation at $600 \times g$ for 5 min and resuspended in DMEM growth medium. The cells were counted, seeded at 8600 cells/cm$^2$ onto 24 × 24 mm glass cover slides and cultured in growth medium supplemented with 0.5 μM flavin adenine dinucleotide (FAD, Carl Roth, # 6833.2), with and without 4 mM TbCl$_3$, either in the dark or under blue light (20.4 μW/cm$^2$) for 4 h at 37 °C and 5% CO$_2$. Then, the cells were washed with PBS and fixed with 4% paraformaldehyde (PFA) in PBS for 10 min at room

temperature. Next, the cells were washed three times with PBS for 10 min, permeabilized with 0.1% Triton-X-100 in PBS for 5 min and washed twice with PBS. The nuclei and actin cytoskeleton were stained with 1 μg/mL Hoechst (Thermo Fisher Scientific, # H3570) and 0.1 μg/mL Phalloidin-iFlour 594 (Abcam, #ab176757) in PBS for 60 min at room temperature. Subsequently, the cells were mounted with Fluoromount-GTM (Thermo Fisher Scientific, #00-4958-02) and fluorescence images were acquired in both channels through a 10× air objective for an area of 1 cm$^2$ by using the tile scan function on an inverted fluorescence microscope (Leica, DMi8). Cell clustering was analyzed by using previously established image analysis tools[27]. In short, the number of cells was quantified by counting the number of nuclei/object in the Hoechst channel and the area of cell clusters by measuring the cell cluster sizes in the actin channel (area >500 μm$^2$ were considered cells, area >10,000 μm$^2$ were considered clusters) with the particle analysis tool in ImageJ version 1.53f51.

## Light-dependent cell aggregation in suspension cultures

Cells were detached from the flask as described earlier, resuspended at $5 \times 10^4$ cell/mL in HBSS supplemented with 1% bovine serum albumin (BSA), 2 mM Ca$^{2+}$ (if not specified otherwise) or 2 mM Tb$^{3+}$, and 0.5 μM FAD. Aliquots of 1 mL were added to 1.5 mL LoBind microfuge tubes (Eppendorf). The cells were incubated on a 3D orbital shaker at 30 rpm at 37 °C for 120 min either under blue light (272 μW/cm$^2$) or in the dark. Subsequently, the cells were transferred to a 12-well plate containing 500 μL of 4% PFA/well. Bright field images were acquired for a total area of 2.5 cm$^2$ through a ×4 air objective. Cell clusters were analyzed in ImageJ by using the particle analysis tool in which objects with a projected area >5000 μm$^2$ (>20 cells) were considered to be clusters (the area for a single cell is 200–250 μm$^2$). The protocol was adapted to light intensities, wavelengths, etc., depending on the parameter in question; for timelines, multiple parallel samples were prepared and single samples analyzed at a given time. To analyse the reversibility of the cell–cell adhesions, we first kept opto-E-cad-MDA cells for 1 or 2 h in the dark and then placed them under blue light for up to 3 h.

## Immunofluorescence

Cells were seeded onto 24 mm × 24 mm glass cover slides placed in a six-well plate at $3 \times 10^5$ cells/well and cultured overnight at 37 °C and 5% CO$_2$ in the presence of 0.5 μM FAD either in the dark or under blue light (20.4 μW/cm$^2$). The next day, the cells were washed with PBS and fixed for 15 min with 4% PFA at room temperature. Subsequently, the cells were washed three times with PBS, blocked with 1% BSA (Sigma-Aldrich # A7030-100G) in PBS for 1 h, and incubated with the primary rabbit-anti-delta-catenin (p120 catenin, cell signalling #59854, diluted 1:800) or the primary mouse-anti-E-cadherin antibody (cell signalling #14472, diluted 1:200) in 1% BSA in PBS overnight at 4 °C. The next day, cells were first washed three times for 10 min with 1% BSA solution, and then incubated for 2 h at room temperature with 1 μg/mL Hoechst dye and the secondary antibody Alexa Fluor 488 goat anti-mouse IgG at 1:1000 dilution (Invitrogen, #A11029) or the anti-rabbit IgG Fab2 Alexa Fluor 647 at 1:1000 dilution (cell signalling #4414S). Following this, the samples were washed three times with PBS for 10 min and mounted with Fluoromount-GTM. Images were acquired by using a confocal microscope (Leica SP8) equipped with a ×63 oil objective. For the staining of cells aggregated for p120, first, the cell aggregation experiments in suspension cultures were performed for 2 h in the dark as described earlier, and the aggregates were allowed to settle for 5 min by gravity. Next, approximately 900 μL of supernatant was removed and the aggregates were mixed with 100 μL of Cultrex Reduced Growth Factor basement membrane extract (BME), type 2 (R&D Systems, # 3533-005-02), with 200 μL tips cut at the end. The mixture was transferred to a μ-Slide eight-well coverslip and incubated at 37 °C for 45 min for the BME gel to solidify. Then, the sample was fixed with 100 μL 4% PFA for 30 min, carefully washed three times with

200 μL PBS for 10 min and blocked with 3% BSA in PBS for 1 h. The sample was incubated with the primary rabbit-anti-delta-catenin delta-1 antibody (1:800 dilution) in 200 μL PBS with 1% BSA overnight at 4 °C. The next day, the sample was washed three times with 1% BSA in PBS, and then incubated with 1 μg/mL Hoechst stain and Alexa Fluor 488 goat anti-mouse IgG secondary antibody (1:1000 dilution) in PBS with 1% BSA for 4 h at room temperature. Finally, the sample was washed three times with 1% BSA in PBS and imaged by using confocal microscopy (Leica SP8).

## Western blot
Cells were cultured to 90% confluence in a sterile square petri dish (Greiner Bio One, 120 × 120 × 17 mm, #688161). The cells were then cultured in the presence of 0.5 μM FAD overnight in the dark or under blue light (20.4 μW/cm$^2$) at 37 °C and 5% $CO_2$. The next day, the cells were washed three times with ice cold PBS, scraped, collected and centrifuged (600 × $g$) at 4 °C, and 1 × 10$^6$ cells were lysed in 50 μL of 1× RIPA buffer (Sigma-Aldrich, #R0278-50ML), 1× protease inhibitor cocktail (Sigma-Aldrich, #SRE0055-1BO) and 5 mM EDTA for 30 min under constant agitation on the vortex mixer with brief sonication steps in between. The protein concentration was quantified by using the Pierce Coomassie (Bradford) Protein assay Kit (Thermo Fisher Scientific, #23200). The samples were run on a 10% Bis-Tris SDS-PAGE gel at 100 V for 20–30 min until samples were compressed along the stacking/resolving interface, and then run at 200 V for 45 min The proteins were blotted onto an activated nitrocellulose membrane (Carl Roth, Transfer membrane Amersham Protran NC 0.45 roll, 400 × 30 cm, #4675.1) by using the Xcell II Blot Module (Thermo Fisher Scientific, # EI9051) at 25 V for 90 min Subsequently, the membrane was blocked with 3% BSA in TBST (50 mM Tris HCl, 150 mM NaCl, 0.1% Tween 20, pH 7.5) for 1 h, and then incubated with the primary antibodies in 1% BSA in TBST at 4 °C overnight. Primary antibodies used were mouse-anti-E-cadherin antibody (cell signalling # 14472; 1:1000 dilution), rabbit anti-catenin-delta-1 (p120, cell signalling #59854; 1:1000 dilution) and mouse-anti-β-actin (Abcam, #ab49900, 1:25,000 dilution). The membrane was then washed three times with TBST and incubated with the secondary antibody (anti-mouse IgG, HRP-linked antibody, cell signalling, #7076, 1:1000 dilution) and anti-rabbit (anti-rabbit IgG, HRP-linked antibody, cell signalling, #7074, 1:1000 dilution) at room temperature for 2 h. Western blots were developed by using Pierce ECL Western Blotting Substrate (Thermo Fisher Scientific, #32106).

## RNA isolation and RT-PCR analysis
To quantify the expression of different EMT markers under different illumination, 1.5 × 10$^5$ opto-E-cad-MDA cells were seeded onto 45 mm cell culture dishes in fresh culture medium supplemented with 0.5 μM FAD. The samples were kept either in the dark or under blue light (20.4 μW/cm$^2$) for 24 h at 37 °C and 5% $CO_2$. To isolate RNA, the culture medium was discarded and the cells were washed twice with PBS, before using 200 μL Trizol to lyse the cells. The lysates were transferred into 1.5 mL reaction tubes and incubated for 5 min before adding 40 μL chloroform and shaking them by hand for 15 s. After 3 min incubation, the samples were spun down at 12,000 × $g$, 6 °C for 15 min The aqueous phases at the top were transferred into fresh 1.5 mL tubes containing 100 μL iso-propanol, incubated for 10 min and then spun down at 12,000 × $g$ and 6 °C for 10 min The supernatants were discarded and each pellet was resuspended in 200 μL 75 % ethanol diluted in DEPC-treated water and mixed by vortexing. Afterwards, the samples were spun down at 7500 × $g$ for 5 min, the supernatants discarded and the pellets left to air dry. Lastly, each pellet was resuspended in 30 μL DEPC-treated water and kept on a heat block for 15 min at 60 °C. The RNA concentration was determined using UV–Vis absorbance.

For reverse transcription, 500 ng RNA was used in each sample according to the manufacturer's protocol using the iScript™ cDNA Synthesis Kit (#1708891, Bio-Rad). The synthesized DNA was subsequently used for the RT-PCR analysis (QuantiNova™ SYBR® Green PCR kit, Qiagen), and the reactions were run on an RT-PCR thermocyler (Azure Cielo 6 thermocycler). The primers for GAPDH (Forward Primer: 5′-CCTGCACCACCAACTGCTTA-3′, Reverse primer: 5′-GGCCATC CACAGTCTTCTGAG-3′), SNAI1 ((Forward Primer: 5′-CTCAAGATGCA-CATCCGAAGCCAC-3′, Reverse primer: 5′-GACACATCGGTCAGACCA-GAGCAC-3′) and FN1 (Forward Primer: 5′-GGAGCAAATGGCACC GAGATA-3′, Reverse primer: 5′-GAGCT GCACATGTCTTGGGAAC-3′) were used for the analysis and each sample was analyzed in three technical replicates from three biological replicates.

## Wound healing assay
Cells were cultured to 80% confluence, detached as described earlier, seeded into a 24-well plate with 1.6 × 10$^5$ cells/well and incubated overnight at 37 °C and 5% $CO_2$ in the dark. The next day, a 200 μL pipette tip was slashed across the well bottom to form a wound within the confluent layer of cells. The cells were washed three times with PBS to remove cell debris. They were given fresh culture medium containing 0.5 μM FAD. The cells were placed onto a microscope (DMi8) equipped with a heating and a $CO_2$ chamber for multi-well plates (37 °C bottom, 40 °C top heating, 5% $CO_2$, Ibidi, #10929). After 1 h equilibration under blue light (272 μW/cm$^2$) or in the dark, bright field images were acquired every 10 min for 16 h. For migration assays in the dark, a long pass filter (514/LP edge basic long pass filter, BLP01-514R-25 from AHF Analysentechnik) was placed in front of the white light source. The wound closure rate was determined with an established script in ImageJ[25], and the migration angle and correlation length were obtained by using MATLAB version 7.10 (R2020a) with the PIVlab–particle image velocimetry (PIV) application developed by Dr. William Thielicke and Prof. Eize J. Stamhuis (MATLAB script: supplementary data)[48]. For spatiotemporal control in the wound-healing assay, the cells were prepared as described earlier, and the experiment was conducted on a confocal microscope (Leica SP8) at 37 °C under 5% $CO_2$. Part of the wound was locally illuminated by using the 488 nm laser (5% intensity for 14.5 h) and images were acquired every 5 min.

## Transwell invasion assay
Cells were detached as described earlier and resuspended in FBS-free medium containing 0.5 μM FAD to 5 × 10$^4$ cells/mL. 100 μL of the cell suspension was added to the upper chamber of each Transwell 24-well plate (Transwell Clear Inserts, polyester (PET) membrane, cell growth area of 0.33 cm$^2$, membrane pore size 8 μm (Corning Product Number 3464)), and 600 μL of growth medium with 10% FBS as a chemoattractant was added to the lower chamber (Supplementary Fig. 9a). The cells were cultured in an incubator at 37 °C with 5% $CO_2$ for 4 h in the dark or under blue light (20.4 μW/cm$^2$). Medium was then removed from both the upper and the lower chambers, the chamber was washed twice with PBS, the cells inside the upper chamber were carefully removed with a moistened cotton swab and the cells in the lower chamber were fixed with 4% PFA for 10 min at room temperature. Next, the cells were washed twice with PBS, stained with 1 μg/mL Hoechst for 1 h at room temperature and washed once more. The cells in the bottom chamber were visualized with a confocal microscope (Leica SP8) and quantified with the particle analysis tool in ImageJ.

## Spheroid culture
Cells were grown to 80% confluence in a T25 flask, the growth medium was removed and the cells were washed twice with PBS. CellTracker Green BODIPY (Thermo Fisher Scientific, #C2102) (final concentration 5 μM) in 5 mL serum-free medium was then added to the cells, and the cells were incubated for 45 min in the cell culture incubator. Subsequently, the cells were washed twice with PBS to remove excess dye.

Afterwards, the cells were detached by using Accutase diluted 1:4 in HBSS for 10 min at room temperature, harvested by centrifugation and resuspended in 1 mL cell culture medium and counted. Afterwards, $2 \times 10^4$ cells/mL were suspended in a spheroid working medium (DMEM, supplemented with 0.66% methyl cellulose (Sigma-Aldrich #M7027), 3% FBS and 1% penicillin/streptomycin, 12.5 mM HEPES) supplemented with 0.5 μM FAD. To form spheroids, we seeded $2 \times 10^3$ cells in 100 μL medium into 96-well U-bottom plates (Greiner Bio-One CELLSTAR, # 650185), spun the cells down for 3 min at 200$g$ to collect them at the bottom of the plate and then incubated them at 37 °C with 5% $CO_2$. The volume of the spheroids was determined with an established MATLAB code, the SpheroidSizer 1_0[49].

### Cell invasion of spheroids into collagen gels

Five millilitres of collagen gel was prepared by mixing 2.62 mL 1.902 mg/mL of collagen type I from foetal bovine skin[50] (final concentration 1 mg/mL) with 500 μL 10× PBS, 500 μL FBS and 200 μL 0.5 mM HEPES on ice and then adjusting the pH to 7 with the addition of 1 M NaOH and diluting the solution to 5 mL with water. Spheroids were prepared in the dark as described earlier and about eight spheroids were resuspended in 200 μL of the freshly prepared collagen mixture and transferred into a μ-Slide eight-well coverslip (Ibidi, #80826) by using 1 mL cut tips. After 1 h at 37 °C and 5% $CO_2$, the collagen gels solidified, and 100 μL of culture medium with 0.5 μM of FAD was added on top and the cells cultured under standard conditions. The following day, the medium above the collagen gels was replaced three times every 2 h to equilibrate the nutrient levels within the gels. Cell invasion was monitored with an inverted confocal fluorescence microscope and z-stacks were acquired throughout the sample by using a 10× objective. The spheroid core area and the number of cells invading the collagen matrix were quantified with ImageJ by using the Analyse Particles tool to measure the area. The largest object in the field of view was defined as the core and the number of all other objects was defined as the number of invading cells.

### CAM assay

The CAM assay was performed according to the established assay[51] guidelines of the European Parliament (2010/63/EU) and the council for the protection of animals in research (§14 TierSchVersV). Fertilized chicken eggs obtained from Brinkschulte GmbH (Senden, Germany) were incubated for 72 h at >60% relative humidity and 37 °C. On the third day of development, the eggs were cracked open and the embryos were transferred to sterile plastic dishes (89 × 89 × 25 mm). Up to four dishes, each containing one embryo, were incubated in a glass petri dish (200 × 30 mm) containing 40 mL of deionized water for increased humidity for another 7 days. At day 6 *ex ovo*, 10 mm Teflon rings were placed on the CAM membrane and the next day, the spheroids were inoculated on top. Spheroids were prepared with $2 \times 10^3$ cells/well from opto-E-cad, MDA-MB-231 and MCF-7 cells, as described earlier; cultured in the dark for 48 h at 37 °C; and harvested by using a 200 μL pipette tip cut. The excess methylcellulose medium was removed once the spheroids settled to the bottom of the tube, and the spheroids were carefully washed twice with 1 mL PBS by using a cut tip. Finally, the spheroids were resuspended in growth medium supplemented with 0.5 μM FAD to eight spheroids/100 μL. Four spheroids in 50 μL medium were placed in each Teflon ring on the CAM at day 7 *ex ovo* and incubated at >60% relative humidity and 37 °C either in the dark or under blue light (20.4 μW/cm²), with two CAMs for each condition and about 3–4 Teflon rings on each CAM, done in biological triplicates. After 24 h, cell invasion was documented through a 1× and 10× objective with a stereomicroscope in the fluorescence and bright field channels (Nikon SMZ25, Tokyo, Japan). For confocal laser scanning microscopy, chicken embryos were sacrificed 24 h after inoculation of the CAM with opto-E-cad spheroids. The tissue in the rings was fixed with 4% PFA as described above. Nuclei were counter-stained

5 min with 20 μM Hoechst33342 and washed twice with PBS. The CAM was cut out along the rings and transferred to 24 × 60 mm coverslips for acquisition of z-stacks at ×20 magnification with a confocal laser scanning microscope (Zeiss LSM 800 and Plan-Apochromat 20×/0.8, Oberkochen, Germany).

### Statistics and reproducibility

All graphs were prepared and statistical analyses performed with OriginPro 2020 version 9.7.0.185. Statistical significance was determined with one-way analysis-of-variance statistical tests with Fisher correction when comparing multiple groups and with the Student's t-tests (two-tailed) when comparing two groups. ns represents $p > 0.05$ (no statistical significance). In the box plots, the box is defined by the first and third quartiles of the data, the line in the box represents the median and the whiskers represent the 5th and 95th percentiles with outliers not shown. In the other graphs, the data are presented as the mean ± s.d. All experiments were performed in at least three biological replicates with two technical duplicates.

### Reporting summary

Further information on research design is available in the Nature Portfolio Reporting Summary linked to this article.

## Data availability

The opto-E-cad plasmid is available through Addgene (Addgene plasmid # 203327). All the data generated in this study are available within the article, the figures, supplementary information and source data. Primary imaging data of large size will be made available upon request for research use within 4 weeks for the next 10 years. Source data are provided with this paper.

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

## Acknowledgements

This work was funded by the European Research Council ERC Starting Grant ARTIST (# 757593, S.V.W.). J.A.E. is financially supported by the Deutsche Forschungsgemeinschaft (DFG grants: SFB1009 project A09 and Eb177/17-1). We would like to thank Sophie Loismann for support with FACS measurements and cell sorting, Nina Knubel for graphical designs, Johanna Bergmann and Yuhao Ji for support with AlphaFold and Alletta Schmidt Hederich for technical assistance in the cell culture.

## Author contributions

B.N.M. and S.V.W. designed the experiments; B.N.M. and C.A.R. performed the experiments, analyzed the data and prepared the figures.

S.N., K.B. and J.A.E supported cell invasion assays into in vitro type I collagen gels and in vivo CAMs. M.M. constructed the opto-E-cadherin plasmid. B.M.B. supported B.N.M. for the analysis of the migration assay and the western blots. B.N.M. and S.V.W. wrote the manuscript. All authors read and reviewed the results and approved the final version of the manuscript.

## Funding

## Competing interests
The authors declare no competing interests.
