## [Peer Review file · Nature Communications]

REVIEWER COMMENTS

Reviewer #1 (Remarks to the Author):

The manuscript of Mombo et al. presents an elegant optogenetic tool to reversibly control cell-cell adhesion mediated by E-cadherin. The tool serves as a switch for single and collective migration in vitro and in vivo. While I like very much the design of the tool and in general I find it useful for studies on cell-cell adhesion dynamics, I have two major concerns:

1. Novelty: besides the reversibility, this study is very similar to a previously published work from Ollech et al. (Nat Commun 2020). I would have expected that the authors could make full use of the tool and show some novelty in the regulation of adherens junctions assembly and disassembly.
2. Tool design: the design of the tool making it dependable on light and calcium is interesting, however the EC 1-2 domains of E-cadherins are crucial for the interactions, and the formation of stable bonds. I wonder how the insertion of LOV2 domain affects the conformational switch and how the adhesion strength between adjacent cells is affected. Moreover, the use of blue light for several hours affects cell viability, particularly in vivo, considering in general also the applicability of optogenetic tools.
3. Biological application: I am completely missing the molecular characterization of adherens junctions. In figure 4, it is shown IFF of actin and p120, but cell densities are rather different when comparing dark and blue light. Also, the WB signal of p120 is reduced under blue light, I do not understand how it is possible that the total protein level changes so quickly depending on the illumination. The cytoplasmic pool of p120 should be the same in the cell lysates, and only the amount localized at AJ should vary. The results on collective migration shown in Fig 5a seem again to suffer from having a different number of cells confined for the wound healing experiment (cells in the dark are more dense than cells exposed to blue light). This affects significantly the collective migration behavior of cell collectives.

Reviewer #2 (Remarks to the Author):

Manuscript: Opto-E-cadherin: Optogenetic control over cell-cell adhesion

The manuscript presented from Mombo and colleagues describes an optogenetic tool capable for the control of cell-cell adhesion experiments at the molecular, cellular and behavioral level in order to study the dynamics and spatiotemporal control of E-cadherin in biological processes. The authors could show

a reversible control of E-cadherin mediated cell-cell adhesion with opto-E-cadherin during blue light exposure through photoregulated calcium binding. They could show subsequent effects on F-actin and p120, effects on cell migration and invasion in 2D and 3D spheroid cultures and in ovo depending on the light (dark or blue light).

The manuscript is well written with clear and detailed figures presenting the results. The authors provided enough detail in the methods part of the manuscript and the interpretation is in line with the results. Taken together the work is of highly interest to the field.

Nevertheless, I have some questions and recommendations for the authors:

1) The authors investigated their opto-E-cadherin in MDA-MB-231 cells and used MCF-7 cells as controls in several experiments. Therefore, the authors proved the functionality in several single cell clones of one tumor entity – it would be good to know if it works as well in different cells (2nd cell line).

2) The authors describe the importance of EMT in relation to reduction or dysfunction of E-Cadherin in carcinomas and other epithelial tissues. Thus, it would be interesting to compare the expression patterns of at least some EMT markers of the opto-E-cad-MDA cells grown in the dark or under blue light in comparison to MDA-MB-231 cells.

3) Results shown in Figure 6: (a) it would be interesting to know if opto-E-cad-MDA cells previously grown in the dark (forming stable spheroids) dissociate upon blue light exposure. (b) How do the authors explain the significant difference between the opto-E-cad-MDA cells under blue light and the MDA-MB-231 cell in the dark? In the other experiments, they mostly behave the same.

4) The authors described that the opto-E-cad-MDA cells no longer formed spheroids and invaded in the CAM under blue light. They showed pictures of the CAM tissue 24 h after seeding the spheroids onto the CAM where we can see cell spreading upon blue light exposure (Figure 6f and Supplementary Figure 9b+c). In my opinion the authors should show the invasion of the cells in sections of the CAM tissue (cryo or IHC), otherwise they cannot see if the cells have invaded the CAM - they only see that the spheroids lose their compact cell shape and spread out as individual cells on the CAM. As the CAM is a very moistly tissue it is possible that the cells are just laying on top of the CAM without invading into the tissue.

In addition, the authors should write in the methods section how many eggs they used for each experimental group and it would be beneficial if the authors would quantify the CAM results as they did it for the in vitro experiments shown in Figure 6.

Reviewer #3 (Remarks to the Author):

The manuscript „Opto-E-cadherin: Optogenetic control over cell-cell adhesions“ by Brice Nzigou Mombo, Brent M. Bijonowski, Stephan Niland, Katrin Brockhaus, Marc Müller, Johannes A. Eble, and Seraphine V. Wegner describes an interesting optogenetic tool to control cell-cell adhesion. To create the novel tool, the authors were integrating a LOV2 domain in epithelial-cadherin (E-cadherin) in a way that the structural changes of the LOV2 domain upon illumination interfere with binding of a calcium ion that is essential for E-cadherin function. The authors did a very good job in characterizing the optogenetic tool. Overall, I agree with the assumption of the authors that the Opto-E-cadherin is a powerful tool to investigate dynamics and spatiotemporal control of biological processes in which E-cadherin is involved and that the novel tool is an improvement over other constructs or methods that aim to introduce switchability in E-cadherins. However, some issues remain that should be addressed before publication.

Major points:

1. Did the authors try different LOV2 domains with mutations that alter the photocycle or show improved behavior? The iLID variant of AsLOV2 and the photocycle mutations V416I, V416L, V416T, and I427T result often in improvements. The switching ratio of the tool in its current state is moderate and might need to be improved to make full use of the tool. Another aspect is the illumination strength that is necessary for full activation. The mutations V416I and V416L usually result in higher light-sensitivity and allow full activation with less light. I would suggest testing these mutations given the slight phototoxicity that is observed in the cell viability assays (see next point).
2. Supplementary Figure 3: Please add statistical analysis to the phototoxicity tests. It looks like that in Figure S3a the opto-E-cad cells have lower viability compared to the control, in other assays not. Please explain.
3. Please add information how the cofactor FMN is available to the LOV2 domain. Is it coming from the growth medium or present during protein synthesis at the endoplasmic reticulum?
4. Page 6 paragraph 4 as well as page 9 until the end of the results & discussion part: please add more information and discussion, why opto-e-cad cells show reversibility of cell-cell adhesion in one assay but not in another?
5. Page 8, end of first paragraph and Figure 5d: I do not agree with the authors that the directionality of the migrating cells is completely lost under blue light. Compared to the cells in darkness more cells migrate in directions between 90 and 180 degrees, however a majority of cells has a directionality between 0 and 90 degrees. Please explain.
6. Supplementary Figure 4, beginning of page 6: the functionality of the opto-e-cad tool is strongly affected by the expression level of the tool. Please add information about the natural expression levels

of e-cadherin in different cell lines and add some discussion how the usage of the opto-e-cad tool might be affected by the dependence on a very strong expression level.

Minor points:

1. Page 3, first paragraph of results: please give a specific information which LOV2 domain fragment was inserted by adding the corresponding amino acid numbers of the beginning and the end of the fragment in the *Avena sativa* phototropin.
2. Page 8 first paragraph: In comparison, the parent...direction of migration (Fig. 5d). It is not really clear to which cell line the authors refer to in the second sentence, if they refer to the parent cells or the opto-E-cad-MDA cells.

Reviewer #1 (Remarks to the Author):

Comment

The manuscript of Mombo et al. presents an elegant optogenetic tool to reversibly control cell-cell adhesion mediated by E-cadherin. The tool serves as a switch for single and collective migration in vitro and in vivo. While I like very much the design of the tool and in general I find it useful for studies on cell-cell adhesion dynamics, I have two major concerns:

Response:

We thank the reviewer for this positive assessment and address all the concerns below.

Comment:

1. Novelty: besides the reversibility, this study is very similar to a previously published work from Ollech et al. (Nat Commun 2020). I would have expected that the authors could make full use of the tool and show some novelty in the regulation of adherens junctions assembly and disassembly.

Response:

The optogenetic tool opto-E-cad has several advantages over the previously published chemo-optogenetic system from Ollech et al., which is mentioned in the manuscript. To recapitulate, in the chemo-optogenetic system a split version of E-cadherin is held together by a photocleavable linker, such that a dissociation of the two domains can be triggered with UV light illumination.

Bidirectional vs. unidirectional switching: The chemo-optogenetic system only allows the adherens junctions to be switched off only once and is therefore irreversible. In contrast to this, the opto-E-cad adhesions can be repeatedly switched on and off in the dark with blue light illumination, respectively and therefore have a wider range of possible uses. The newly added **Fig. 3c**, where cell-cell adhesions form when illumination is stopped and **Fig. 3d**, where adhesions disassemble upon blue light illumination more clearly demonstrates the bidirectional switching capabilities. This aspect is also described in the main text:

“To demonstrate that opto-E-cad can be used to induce cell-cell adhesions and reverse them as desired, we first kept opto-E-cad-MDA cells under blue light for 60 min and then placed them in the dark. Here, we observed an increase in cell aggregation only for cells after turning the illumination off and cells kept under illumination remained as single cells (Fig. 3c).”

Three vs. single component system: The opto-E-cad is significantly simpler and thus easier to use, requiring only one component, while the chemo-optogenetic system consists of three components. The chemo-optogenetic system requires two transfected E-cadherin components and the addition of a custom synthesized small molecule, which all have to be matched in concertation for the system to work. In contrast, the opto-E-cad only requires one protein to be transfected. Therefore, the opto-E-cad is easier to implement in other studies and will be made available via Addgene.

UV light vs. blue light: The opto-E-cad responds to low intensities of blue light, which are not toxic to the cell as has been shown in this (**Supplementary Fig. 4 a-c**) and numerous other optogenetic studies with the LOV2 domain. All of these points are summarized in the manuscript:

“The opto-E-cad with a photoswitchable AsLOV2 domain has several advantages over existing systems that involve light-responsive small molecules or optogenetic systems that rely on light-dependent dimerization. The fact that the opto-E-cad is a one-component optogenetic tool avoids all the problems that can arise from two or more component systems where the expression levels of different constructs have to be matched. The LOV2 domain can undergo repeated conformational changes over many blue

light/dark cycles, which allows one to dynamically turn the opto-E-cad adhesions off and on again without using cell-toxic UV light.”

Comment:

2. Tool design: the design of the tool making it dependable on light and calcium is interesting, however the EC 1-2 domains of E-cadherins are crucial for the interactions, and the formation of stable bonds. I wonder how the insertion of LOV2 domain affects the conformational switch and how the adhesion strength between adjacent cells is affected. Moreover, the use of blue light for several hours affects cell viability, particularly *in vivo*, considering in general also the applicability of optogenetic tools.

Response:

The point of how the insertion of the LOV2 domain affects the binding between E-cadherins is justified. For this purpose, we predicted the structure of opto-E-cad using AlphaFold and overlaid it with the known *cis*- (PDB: 4ZT1) and *trans*- (PDB: 2O72) dimer structures of human E-cadherin (**Supplementary Fig. 1**). Both in the *cis*- and the *trans*-dimer the LOV2 domain is in a location away from the dimerization interface in a way that the interaction between the EC 1-2 domains of E-cadherin are not affected. We added following text to the manuscript to highlight this point.

*“When inserting the LOV2 domain at the chosen position, we further paid attention not to sterically block homophilic *cis*- and *trans*-interactions between E-cadherins by modelling the opto-E-cad structure with AlphaFold and aligning it with the crystal structures of the E-cadherin dimers (**Supplementary Fig. 1b,c**). In both cases, the LOV2 domain is oriented in a way that does not affect the interaction interface.”*

We have investigated the issue of light toxicity in different experimental setups (**Supplementary Fig. 4a-c**). Moreover, negative control experiments with the parent cell line MDA-MB-231 in the dark and under blue light showed no effect of illumination on cell behavior including cell clustering in 2D (**Fig. 1c**, **Supplementary Fig. 3b**) and 3D (**Fig. 2b**, **Supplementary Fig. 3e**) cultures, cell migration rate in the wound healing assay (**Fig. 5c**), spheroid compaction (**Fig. 6b**), cell invasion into collagen hydrogels (**Fig. 6d-e**, **Supplementary Fig. 11a**) and the CAM assay (**Supplementary Fig. 11c**). Overall, we can conclude that the illumination with blue light has no effect on cell viability and the different cell behavior here investigated. The numerous optogenetic studies including ones *in vivo* over long periods have well established that the required intensities of blue light are not toxic.

Comment:

3. Biological application: I am completely missing the molecular characterization of adherens junctions. In figure 4, it is shown IFF of actin and p120, but cell densities are rather different when comparing dark and blue light. Also, the WB signal of p120 is reduced under blue light, I do not understand how it is possible that the total protein level changes so quickly depending on the illumination. The cytoplasmic pool of p120 should be the same in the cell lysates, and only the amount localized at AJ should vary. The results on collective migration shown in Fig 5a seem again to suffer from having a different number of cells confined for the wound healing experiment (cells in the dark are more dense than cells exposed to blue light). This affects significantly the collective migration behavior of cell collectives.

Response:

Characterization of the signaling: For better molecular characterization of the E-cadherin associated cellular signaling, we investigated the up regulation of EMT markers under blue light illumination using RT-PCR (**Fig. 4e**). In these new experiments, the EMT markers Snai1 and fibronectin (FN1) were upregulated under blue light illumination compared to the dark samples.

“Finally, we investigated if photoregulation of the opto-E-cad alters E-cadherin associated cellular signalling. For this purpose, we measured the upregulation of EMT markers in opto-E-cad-MDA cells after overnight incubation under blue light illumination using RT-PCR (Fig. 4e). The mRNA levels for both the EMT master regulator Snai1 and the extracellular matrix protein fibronectin-1 increased under blue light illumination compared to cells kept in the dark. Overall, these results show that the opto-E-cad allows to switch to a more mesenchymal phenotype under blue light illumination compared to the dark by changing the connection to the actin cytoskeleton, the intracellular interactions with catenins and the gene expression profile.”

Cell density: We thank the reviewer for pointing out that the apparent differences in cell density in the actin staining and IFF images (Fig. 4) and the wound healing assay (Fig. 5a) in the dark and under blue light misleading. In the dark, the cells grow in groups leading to areas of higher and lower local cell density and under blue light illumination, the cell distribution is uniform (large area scan, Supplementary Fig. 3b). Moreover, the cell-cell adhesions in the dark lead to changes in cell morphology and a smaller spreading area than under blue light, which is why cell densities in the dark samples appear higher than under blue light. By scanning large areas of the samples (1 cm²) and counting the number of nuclei after Hoechst staining, we verified that the total cell numbers in the dark and blue light samples are the same (Supplementary Fig. 3a). Likewise, cell viability assays show that the samples in the dark and under blue light have similar cell number (Supplementary Fig. 4).

WB of p120: These assays were conducted after incubating the opto-E-cad-MDA cells in the dark or under blue light overnight, and the observed differences are not on a fast time scale. The association of p120 to the intracellular domain of E-cadherin increases its overall stability. Therefore, it is not just the localization but also the overall levels of p120 that change depending on its binding to E-cadherin. A similar decrease in p120 concentration with decreased E-cadherin expression has also been reported in other studies (Law *et al.* Oncogene (2013) 32, 1316–1329; Soto *et al.* J. Cell Biol. (2008) 183 (4): 737–749).

Reviewer #2 (Remarks to the Author):

Comment:

The manuscript presented from Mombo and colleagues describes an optogenetic tool capable for the control of cell-cell adhesion experiments at the molecular, cellular and behavioral level in order to study the dynamics and spatiotemporal control of E-cadherin in biological processes. The authors could show a reversible control of E-cadherin mediated cell-cell adhesion with opto-E-cadherin during blue light exposure through photoregulated calcium binding. They could show subsequent effects on F-actin and p120, effects on cell migration and invasion in 2D and 3D spheroid cultures and in ovo depending on the light (dark or blue light).

The manuscript is well written with clear and detailed figures presenting the results. The authors provided enough detail in the methods part of the manuscript and the interpretation is in line with the results. Taken together the work is of highly interest to the field.

Response:

We thank the reviewer for the positive assessment of our work. We have addressed all the points raised by the reviewer as detailed below.

Comment:

Nevertheless, I have some questions and recommendations for the authors:

1) The authors investigated their opto-E-cadherin in MDA-MB-231 cells and used MCF-7 cells as controls

in several experiments. Therefore, the authors proved the functionality in several single cell clones of one tumor entity – it would be good to know if it works as well in different cells (2nd cell line).

Response:

We have introduced the opto-E-cadherin into other cell types including HeLa, L929 and A431D cells with a specific E-cadherin knockout and established stable opto-E-cad cell lines from them. In all of these opto-E-cad cell lines, we observed higher cell aggregation in the dark than under blue light (and the parent cell line) (**Supplementary Fig. 6**). Moreover, we added the spheroids of opto-E-cad-A431D cells to the manuscript as a second demonstration of photoswitchable cell-cell adhesions in 3D cell culture (**Supplementary Fig. 10**). Also in this case we observed more compact spheroids in the dark than under blue light illumination. In the main text we have inserted this information:

*“Similarly, also spheroids formed from opto-E-cad-A431D cells were also smaller in volume when kept in the dark than under blue light (**Supplementary Fig. 10**).”*

Comment:

2) The authors describe the importance of EMT in relation to reduction or dysfunction of E-Cadherin in carcinomas and other epithelial tissues. Thus, it would be interesting to compare the expression patterns of at least some EMT markers of the opto-E-cad-MDA cells grown in the dark or under blue light in comparison to MDA-MB-231 cells.

Response:

Following the suggestion of the reviewer, we conducted RT-PCR experiments and measured the expression of EMT markers –Snai1 and fibronectin - in opto-E-cad-MDA cells after overnight incubation in the dark or under blue light (**Fig. 4e**). For both mesenchymal markers, we observed higher mRNA level under blue light than in the dark. We have added following text to the manuscripts:

*“Finally, we investigated if photoregulation of the opto-E-cad alters E-cadherin associated cellular signalling. For this purpose, we measured the upregulation of EMT markers in opto-E-cad-MDA cells after overnight incubation under blue light illumination using RT-PCR (**Fig. 4e**). The mRNA levels for both the EMT master regulator Snai1 and the extracellular matrix protein fibronectin-1 increased under blue light illumination compared to cells kept in the dark. Overall, these results show that the opto-E-cad allows switching to a more mesenchymal phenotype under blue light illumination compared to darkness, changing the connection to the actin cytoskeleton, the intracellular interactions with catenins and the gene expression profile.”*

Comment:

3) Results shown in Figure 6: (a) it would be interesting to know if opto-E-cad-MDA cells previously grown in the dark (forming stable spheroids) dissociate upon blue light exposure. (b) How do the authors explain the significant difference between the opto-E-cad-MDA cells under blue light and the MDA-MB-231 cell in the dark? In the other experiments, they mostly behave the same.

Response:

(a) In this experiment, the spheroids were formed in methylcellulose media, which does not have any adhesion motif for cell-matrix interactions. Therefore, once the opto-E-cad-MDA spheroids have compacted in the dark (where cell-cell adhesions form), the spheroids will not dissociate again even if placed under blue light. This lack of dissociation is not necessarily due to a lack of reversion of the opto-E-cad but because the cells are in a non-adhesive extracellular environment and there is no driving force for the cells to move away from each other. This is unlike the scenario in the collagen matrix (**Fig. 6c**) or the CAM assay (**Fig. 6f**), there the cells are placed in an adhesive ECM and under blue light invade into

the surrounding as the cell-cell adhesions are weak and the cell-matrix adhesions are strong. We have clarified this point in the revised manuscript.

“In this experimental set-up, the opto-E-cad mediated adhesions were reversed after 2 days when the spheroids were placed in the collagen gels. In this case, the strong cell-matrix adhesions may support the cells to overcome residual cell-cell adhesions and invade into the matrix as single cells.”

(b) The significant differences between opto-E-cad-MDA cells under blue light and the MDA-MB-231 cells in the dark indicate that there is still some residual attraction between opto-E-cad-MDA cells under blue light. We commented on this in the revised manuscript.

“The higher compactness of opto-E-cad-MDA spheroids under blue light than MDA-MB-231 spheroids indicate that even in the dark that there is some residual attraction between opto-E-cad-MDA cells. As no switching is complete some background activity is always expected and this background may contribute differently in various assays depending on the sensitivity. For example, in the 3D clustering assay the cells are mildly agitated at 30 rpm, which may already rupture weak interactions. In contrast, for spheroid formation, the cells are centrifuged to the bottom of the U-well and kept still in the incubator in a non-adhesive environment. In the absence of other forces (e.g. cell-matrix, shear forces) even weak cell-cell interaction lead to some spheroid compacting.”

Comment:

4) The authors described that the opto-E-cad-MDA cells no longer formed spheroids and invaded in the CAM under blue light. They showed pictures of the CAM tissue 24 h after seeding the spheroids onto the CAM where we can see cell spreading upon blue light exposure (Figure 6f and Supplementary Figure 9b+c). In my opinion the authors should show the invasion of the cells in sections of the CAM tissue (cryo or IHC), otherwise they cannot see if the cells have invaded the CAM - they only see that the spheroids lose their compact cell shape and spread out as individual cells on the CAM. As the CAM is a very moistly tissue it is possible that the cells are just laying on top of the CAM without invading into the tissue.

In addition, the authors should write in the methods section how many eggs they used for each experimental group and it would be beneficial if the authors would quantify the CAM results as they did it for the in vitro experiments shown in Figure 6.

Response:

We thank the reviewer for this excellent suggestion and indeed see the need to differentiate between opto-E-cad cells just dissociating from each other under blue light and opto-E-cad cells invading into the CAM/interacting with cells in the developing embryo under blue light. For this purpose, we repeated the CAM assay with stained opto-E-cad-MDA cells (CellTracker Green). At the end of the experiment fixed the cells and stained all nuclei with Hoechst (**Fig. 6g**). 3D confocal image stacks of opto-E-cad-MDA cells in the CAM assay show that in the dark the spheroids only sat on top of the CAM, whereas under blue light the opto-E-cad-MDA cells invaded the CAM and mixed with its cells. The new Figure **Fig. 6g** clearly shows that the opto-E-Cad cells have penetrated the entire mesenchymal layer of the CAM to the allantoic endoderm.

We added the number of CAMs and replicates to the experimental section. We find it difficult to quantify adequately the observations in the CAM assay as the CAM leads to a high background in the epifluorescence and makes it difficult to detect single cells that invade into the CAM with automated image analysis. We added to the main text of the manuscript:

*“In confocal fluorescence images of samples where all nuclei were stained (shown in blue), the spheroids only loosely attached to the chorionic ectodermal surface of the CAM (filled arrow heads) in the dark with no cells detaching from spheroids and infiltrating the CAM (**Fig. 6g**). In contrast under blue light, the*

cells invaded the entire CAM all the way down to its allantoic endoderm side (open arrow heads) and opto-E-cad-MDA cells intermixed with the surrounding cells.”

“Four spheroids in 50 μ L medium were placed in each Teflon ring on the CAM at day 7 ex ovo and incubated at >60% relative humidity and 37 °C either in the dark or under blue light (20.4 μ W/cm²), with two CAMs for each condition and about 3-4 Teflon rings on each CAM, done in biological triplicates.”

“For confocal laser scanning microscopy, chicken embryos were sacrificed 24 h after inoculation of the CAM with opto-E-cad spheroids. The tissue in the rings was fixed with 4% PFA as described above. Nuclei were counter-stained 5 min with 20 μ M Hoechst33342 and washed twice with PBS. The CAM was cut out along the rings and transferred to 24 \times 60 mm coverslips for acquisition of z-stacks at 20 \times magnification with a confocal laser scanning microscope (Zeiss LSM 800 and Plan-Apochromat 20x/0.8, Oberkochen, Germany).”

Reviewer #3 (Remarks to the Author):

Comment:

The manuscript „Opto-E-cadherin: Optogenetic control over cell-cell adhesions“ by Brice Nzigou Mombo, Brent M. Bijonowski, Stephan Niland, Katrin Brockhaus, Marc Müller, Johannes A. Eble, and Seraphine V. Wegner describes an interesting optogenetic tool to control cell-cell adhesion. To create the novel tool, the authors were integrating a LOV2 domain in epithelial-cadherin (E-cadherin) in a way that the structural changes of the LOV2 domain upon illumination interfere with binding of a calcium ion that is essential for E-cadherin function. The authors did a very good job in characterizing the optogenetic tool. Overall, I agree with the assumption of the authors that the Opto-E-cadherin is a powerful tool to investigate dynamics and spatiotemporal control of biological processes in which E-cadherin is involved and that the novel tool is an improvement over other constructs or methods that aim to introduce switchability in E-cadherins. However, some issues remain that should be addressed before publication.

Response:

We thank the reviewer for the positive assessment and the questions raised that we address below.

Comment:

Major points:

1. Did the authors try different LOV2 domains with mutations that alter the photocycle or show improved behavior? The iLID variant of AsLOV2 and the photocycle mutations V416I, V416L, V416T, and I427T result often in improvements. The switching ratio of the tool in its current state is moderate and might need to be improved to make full use of the tool. Another aspect is the illumination strength that is necessary for full activation. The mutations V416I and V416L usually result in higher light-sensitivity and allow full activation with less light. I would suggest testing these mutations given the slight phototoxicity that is observed in the cell viability assays (see next point).

Response:

The optimization of the opto-E-cad concerning different parameters of merit (dark/light switching efficiency, altered photocycle times, higher light sensitivity) is certainly desirable. Unfortunately, the here described experimental setups are not suitable for high throughput screening of different constructs as this would involve the generation of monoclonal stable cell lines for each mutation and find clones with a similar expression levels for comparison. Such experiments probably would be more feasible by setting up a screening method based on purified proteins in a follow up study.

Comment:

2. Supplementary Figure 3: Please add statistical analysis to the phototoxicity tests. It looks like that in Figure S3a the opto-E-cad cells have lower viability compared to the control, in other assays not. Please explain.

Response:

After the statistical analysis of the phototoxicity measurements, we see that there is no significant impact on cell viability (**Supplementary Fig. 4**). Moreover, negative control experiments with the parent cell line MDA-MB-231 in the dark and under blue light showed no effect of illumination on cell behavior including cell clustering in 2D (**Fig. 1c, Supplementary Fig. 3b**) and 3D (**Fig. 2b, Supplementary Fig. 3e**) cultures, cell migration rate in the wound healing assay (**Fig. 5c**), spheroid compaction (**Fig. 6b**), cell invasion into collagen hydrogels (**Fig. 6d-e, Supplementary Fig. 11a**) and in the CAM assay (**Supplementary Fig. 11c**). Overall, we can conclude that the illumination with blue light has no effect on cell viability and the different cell behavior investigated here.

Comment:

3. Please add information how the cofactor FMN is available to the LOV2 domain. Is it coming from the growth medium or present during protein synthesis at the endoplasmic reticulum?

Response:

We added the cofactor to the medium externally in all experiments and have now added data to show that the addition of the cofactor is essential for the photoresponse of opto-E-cad (**Supplementary Fig. 5c**).

“The integration of the cofactor is essential for the function of the LOV2 domain in the opto-E-cad and we found that the addition of the cofactor to the culture media was essential for the function (Supplementary Fig. 5c).”

Comment:

4. Page 6 paragraph 4as well as page 9 until the end of the results & discussion part: please add more information and discussion, why opto-e-cad cells show reversibility of cell-cell adhesion in one assay but not in another?

Response:

The apparent differences in reversibility from one assay to the other is connected to the time scale and other forces (cell-matrix or shear forces) present in the different systems. To clarify these points we added following text to the manuscript.

“In this experimental set-up, the opto-E-cad mediated adhesions were reversed after 2 days when the spheroids were placed in the collagen gels. In this case, the strong cell-matrix adhesions may support the cells to overcome residual cell-cell adhesions and invade into the matrix as single cells.”

“The higher compactness of opto-E-cad-MDA spheroids under blue light than MDA-MB-231 spheroids indicate that even in the dark that there is some residual attraction between opto-E-cad-MDA cells. As no switching is complete some background activity is always expected and this background may contribute differently in various assays depending on the sensitivity. For example, in the 3D clustering assay the cells are mildly agitated at 30 rpm, which may already rupture weak interactions. In contrast, for spheroid formation, the cells are centrifuged to the bottom of the U-well and kept still in the incubator in a non-adhesive environment. In the absence of other forces (e.g. cell-matrix, shear forces) even weak cell-cell interaction lead to some spheroid compacting.”

Comment:

5. Page 8, end of first paragraph and Figure 5d: I do not agree with the authors that the directionality of the migrating cells is completely lost under blue light. Compared to the cells in darkness more cells migrate in directions between 90 and 180 degrees, however a majority of cells has a directionality between 0 and 90 degrees. Please explain.

Response:

We have moderated our claim in the text.

“Similarly, in the dark, the migration angles of the opto-E-cad-MDA cells were between 0 and 90 degrees, showing movement directly into the wound, whereas under blue light, a smaller fraction of cells had a migration angle between 0 and 90 degrees, showing less coordination in the migration (Fig. 5d).”

Comment:

6. Supplementary Figure 4, beginning of page 6: the functionality of the opto-e-cad tool is strongly affected by the expression level of the tool. Please add information about the natural expression levels of e-cadherin in different cell lines and add some discussion how the usage of the opto-e-cad tool might be affected by the dependence on a very strong expression level.

Response:

For E-cadherin to have an impact on cell morphology, clustering and behavior a critical number of molecules per cell have to be present. Therefore, it is expected for low opto-E-cadherin level not to result in a change in cell aggregation. For the expression of E-cadherin in different cell lines usually relative expression levels are determined by western blot or flow cytometry and unfortunately there are few studies that report absolute values. For example, Lombaerts *et al.* Br. J. Cancer 2006, 94, 661–671 reports the qualitative differences E-cadherin expression for MCF-7 and MDA-Mb-231 cells (any many other cell types) by western blot and found the expression to be 47 vs. 18 MESF, respectively, by flow cytometry. Silvestre *et al.* (Langmuir 2009, 25, 17, 10092–10099) measured E-cadherin surface densities by FACS and fluorescent calibration beads on wild type MDA-MB231 cells and on MDA-MB231 cells engineered to express E-cadherin under tetracycline control. They report 18-46 cadherins per μm^2 (ca. 10^5 molecules per cell) in MDA-MB-231 cells with a doxycycline inducible E-cadherin plasmids, which is similar to the levels in this study. We recognize that the opto-E-cad construct will be most useful in a low endogenous E-cadherin background and have added following comment to the manuscript:

“All of the here tested cell lines have low native E-cadherin expression and the opto-E-cadherin is best used in cells with a low E-cadherin background or specific E-cadherin knockouts.”

Comment:

Minor points:

1. Page 3, first paragraph of results: please give a specific information which LOV2 domain fragment was inserted by adding the corresponding amino acid numbers of the beginning and the end of the fragment in the Avena sativa phototropin.

Response:

We included the corresponding amino acid numbers of the AsLOV2 fragment we used to the text and the complete sequence is in the supporting information.

“Part of this loop are the side chains of D134 and D136 and the carbonyl oxygen of N143, which are ligands for two of the three Ca^{2+} ions between EC1 and EC2. The LOV2 domain of Avena sativa phototropin 1 (AsLOV2, 404-542) inserted before D134 has a well-folded Per-Arnt-Sim (PAS) domain and a flavin mononucleotide (FMN) chromophore cofactor.”

Comment:

2. Page 8 first paragraph: In comparison, the parent...direction of migration (Fig. 5d). It is not really clear to which cell line the authors refer to in the second sentence, if they refer to the parent cells or the opto-E-cad-MDA cells.

Response:

We clarified in the sentence that we refer to the opto-E-cad-MDA cells.

Reviewers' comments:

Reviewer #2 (Remarks to the Author)

My questions and comments were fully answered and practically implemented by the authors. The replicated experiments (RT PCR of the EMT markers and confocal microscopy of the CAM) support the authors' assumptions. I have no further additions.

Reviewer #3 (Remarks to the Author)

The authors of the manuscript „Opto-E-cadherin: Optogenetic control over cell-cell adhesions“ by Brice Nzigou Mombo, Brent M. Bijonowski, Stephan Niland, Katrin Brockhaus, Marc Müller, Johannes A. Eble, and Seraphine V. Wegner did a careful revision of their manuscript and successfully addressed most of the issues raised by me. I am not fully convinced that a testing of LOV2 photocycle mutants would not be possible, as they tested their construct in several different cell lines. Testing of known LOV2 photocycle mutants should be the same amount of work and it is not clear why high-throughput screening would be necessary.

As all the other issues have been resolved, I recommend publication.

Reviewer #4 (Remarks to the Author) (replacement reviewer for Reviewer #1):

Reviewer #1 (Remarks to the Author):

Comment:

1. Novelty: besides the reversibility, this study is very similar to a previously published work from Ollech et al. (Nat Commun 2020). I would have expected that the authors could make full use of the tool and show some novelty in the regulation of adherens junctions assembly and disassembly.

The authors' response is based on 3 arguments:

- The reversibility of their approach, while the approach developed by Ollech et al. is irreversible by nature due to the cleavage of the chemical linker used in this approach. This argument is solid and the

addition of new figures 3c and 3d tend to show the claimed reversibility. In fact, these figures indeed show the bidirectionality of the control of cell-cell adhesion which strongly suggests that it is reversible, but it could be further substantiated by showing the effect of alternating pulses of illumination/dark condition in order to really prove this reversibility.

- Three vs. single component system : this is also a solid argument, since simplicity of use proves to be experimentally critical for the success of optogenetic arguments. As argued by the author, stoichiometry is a critical factor for bi-component optogenetic systems and can render their use difficult, although that can be bypassed with FACS-sorting or bi-cistronic expression vectors. In addition, the requirement of a non-commercial chemical (Ha-pl-BG) decreases the usability of Ollech's approach.

- UV light required for the optochemical approach is cytotoxic on the long term and makes it less amenable for long-term experiments

In those respects, the authors' approach seems more easily useful for most usages.

Comment:

2. Tool design: the design of the tool making it dependable on light and calcium is interesting, however the EC 1-2 domains of E-cadherins are crucial for the interactions, and the formation of stable bonds. I wonder how the insertion of LOV2 domain affects the conformational switch and how the adhesion strength between adjacent cells is affected. Moreover, the use of blue light for several hours affects cell viability, particularly in vivo, considering in general also the applicability of optogenetic tools.

To the first point (the effect of LOV2 domain insertion in E-cadherin), the authors respond by referring to structure predictions made with AlphaFold. These predictions indeed show that the LOV2 domain should not affect cis- and trans-dimer formation. However, a proper experimental demonstration would be more satisfactory. While micropipette-based cell-cell adhesion strength (opto-Ecad-MDA vs. Ecad-MDA) or in vitro measurement of adhesion between opto-Ecad vs. E-cad single proteins (through optical tweezer or AFM-based experiments for instance) would be the proper demonstration of the fact that LOV2 doesn't affect E-cadherin adhesive properties, these experiments might prove difficult and not be necessary. However, a more detailed comparison of opto-Ecad-MDA vs. Ecad-MDA morphological and functional properties (in particular the organization of the adherens junctions and the adhesive properties of these cells) would be useful.

To the second point, they respond with an evaluation of toxicity at different light intensities over periods of time up to 24 hours. It appears that cytotoxicity is minimal albeit apparently not null (significant 30% decrease of viability in MDA-MB_231 cells in supp. Fig. 4c, no significant decrease in the other figures) at the highest power tested, which is the one used for most experiments (and rightfully so, since Supp. Fig 5a shows that the effect on adhesion is less pronounced at an illumination intensity twice lower). However, that is true with most optogenetic systems used that are based on blue light-sensitive proteins in their majority and is still probably better than with UV-based systems as used in Ollech et al. The observed cytotoxicity thus appears satisfactory.

Comment:

3. Biological application: I am completely missing the molecular characterization of adherens junctions. In figure 4, it is shown IFF of actin and p120, but cell densities are rather different when comparing dark and blue light. Also, the WB signal of p120 is reduced under blue light, I do not understand how it is possible that the total protein level changes so quickly depending on the illumination. The cytoplasmic pool of p120 should be the same in the cell lysates, and only the amount localized at AJ should vary. The results on collective migration shown in Fig 5a seem again to suffer from having a different number of cells confined for the wound healing experiment (cells in the dark are more dense than cells exposed to blue light). This affects significantly the collective migration behavior of cell collectives.

There are 2 points raised here by reviewer 1:

1) that cell density is different in conditions in the dark or after blue illumination in the experiments of Immunofluorescence (fig 4B) and collective migration (Fig 5a), which could affect the result. It is true, like reviewer 1 says, that cell density has a strong effect on the organization of cell junctions and on cell migration. In that respect, the authors' response is not fully satisfactory. In particular, they claim that total cell numbers are the same in the dark or light samples, but that is another experiment when cells are far from confluency (Supp. Fig 3a), and does not solve the fact that comparison of adherens junctions and migration as presented in Figures 3 and 4 could result from differences in cell densities.

The authors claim that these observed differences in density are due to different cell spreading and clustering in the dark or under blue light illumination. This could be circumvented by growing the cells in dark conditions until confluency and then illuminating them for several hours.

2) that it is surprising that p120 expression decreases so quickly after blue light illumination (Fig. 4d). The authors clarify the experimental conditions, explaining that illumination has been conducted overnight, a timescale compatible with p120 down-regulation. This should be made clearer in the Material and Methods section and in the caption of Fig. 4. In addition, they argue that not only membrane localization but also expression of p120 is affected by E-cadherin expression, and cite several articles to substantiate it. However, it is worth noting that E-cadherin expression is not changed in their experiments after blue light illumination as shown by Fig. 4d, only its adhesive engagement is. How engagement of E-cadherin in adhesive bonds alters p120 stability should be clarified.

Reviewer #2 (Remarks to the Author)

My questions and comments were fully answered and practically implemented by the authors. The replicated experiments (RT PCR of the EMT markers and confocal microscopy of the CAM) support the authors' assumptions. I have no further additions.

Response:

We thank the reviewer for the suggestions and support in improving the quality of the manuscript.

Reviewer #3 (Remarks to the Author)

The authors of the manuscript „Opto-E-cadherin: Optogenetic control over cell-cell adhesions“ by Brice Nzigou Mombo, Brent M. Bijonowski, Stephan Niland, Katrin Brockhaus, Marc Müller, Johannes A. Eble, and Seraphine V. Wegner did a careful revision of their manuscript and successfully addressed most of the issues raised by me. I am not fully convinced that a testing of LOV2 photocycle mutants would not be possible, as they tested their construct in several different cell lines. Testing of known LOV2 photocycle mutants should be the same amount of work and it is not clear why high-throughput screening would be necessary.

As all the other issues have been resolved, I recommend publication.

Response:

The comparison of different LOV2 mutants is technically possible but time intensive as one has to generate stable cell lines with similar expression levels.

We thank the reviewer for the suggestions and support in improving the quality of the manuscript.

Reviewer #4 (Remarks to the Author) (replacement reviewer for Reviewer #1):

Reviewer #1 (Remarks to the Author):

Comment:

1. Novelty: besides the reversibility, this study is very similar to a previously published work from Ollech et al. (Nat Commun 2020). I would have expected that the authors could make full use of the tool and show some novelty in the regulation of adherens junctions assembly and disassembly.

The authors' response is based on 3 arguments:

- The reversibility of their approach, while the approach developed by Ollech et al. is irreversible by nature due to the cleavage of the chemical linker used in this approach. This argument is solid and the addition of new figures 3c and 3d tend to show the claimed reversibility. In fact, these figures indeed show the bidirectionality of the control of cell-cell adhesion which strongly suggests that it is reversible, but it could be further substantiated by showing the effect of alternating pulses of illumination/dark condition in order to really prove this reversibility.

- Three vs. single component system: this is also a solid argument, since simplicity of use proves to be experimentally critical for the success of optogenetic arguments. As argued by the author, stoichiometry is a critical factor for bi-component optogenetic systems and can render their use difficult, although that can be bypassed with FACS-sorting or bi-cistronic expression vectors. In addition, the requirement of a non-commercial chemical (Ha-pl-BG) decreases the usability of Ollech's approach.

- UV light required for the optochemical approach is cytotoxic on the long term and makes it less

amenable for long-term experiments

In those respects, the authors' approach seems more easily useful for most usages.

Response:

We thank the reviewer for agreeing with our argumentation.

Following the reviewer's suggestion, we further validated the reversibility of the opto-E-cad mediated cell-cell adhesions in repeated illumination/dark cycles and added the new data to Figure 3e and added following text to the manuscript.

"The opto-E-cad based cell-cell adhesions could also be switched on and off repeatedly as shown in three dark/blue light cycles with one-hour intervals (Fig. 3e). In these experiments there was no sign of fatigue in the photoswitching."

Comment:

2. Tool design: the design of the tool making it dependable on light and calcium is interesting, however the EC 1-2 domains of E-cadherins are crucial for the interactions, and the formation of stable bonds. I wonder how the insertion of LOV2 domain affects the conformational switch and how the adhesion strength between adjacent cells is affected. Moreover, the use of blue light for several hours affects cell viability, particularly in vivo, considering in general also the applicability of optogenetic tools.

To the first point (the effect of LOV2 domain insertion in E-cadherin), the authors respond by referring to structure predictions made with AlphaFold. These predictions indeed show that the LOV2 domain should not affect cis- and trans-dimer formation. However, a proper experimental demonstration would be more satisfactory. While micropipette-based cell-cell adhesion strength (opto-Ecad-MDA vs. Ecad-MDA) or in vitro measurement of adhesion between opto-Ecad vs. E-cad single proteins (through optical tweezer or AFM-based experiments for instance) would be the proper demonstration of the fact that LOV2 doesn't affect E-cadherin adhesive properties, these experiments might prove difficult and not be necessary. However, a more detailed comparison of opto-Ecad-MDA vs. Ecad-MDA morphological and functional properties (in particular the organization of the adherens junctions and the adhesive properties of these cells) would be useful.

To the second point, they respond with an evaluation of toxicity at different light intensities over periods of time up to 24 hours. It appears that cytotoxicity is minimal albeit apparently not null (significant 30% decrease of viability in MDA-MB_231 cells in supp. Fig. 4c, no significant decrease in the other figures) at the highest power tested, which is the one used for most experiments (and rightfully so, since Supp. Fig 5a shows that the effect on adhesion is less pronounced at an illumination intensity twice lower). However, that is true with most optogenetic systems used that are based on blue light-sensitive proteins in their majority and is still probably better than with UV-based systems as used in Ollech et al. The observed cytotoxicity thus appears satisfactory.

Response:

We also agree that measuring the adhesion forces directly would of course be a good addition to the manuscript but as pointed out by the reviewer requires specialized and complicated techniques. Yet, to show that opto-E-cad based adhesions are mechanically stable enough despite the added LOV2 domain, we looked at the influence on cell morphology (Fig. 4c). We observe that the opto-E-cad cells in the dark that form cell-cell adhesions in the clusters have on average a smaller spreading area than the parent cell line MDA-MB-231 (negative control) and single opto-E-cad cells not forming cell-cell adhesions. Moreover, the opto-E-cad cells (positive control) in clusters have a

similar spreading area as E-cad-MDA cells. The changes in cell morphology together with data that show the p120 recruitment to opto-E-cad based adhesion and the changes in the actin cytoskeleton in the dark all indicate that opto-E-cad has sufficient mechanical stability to mediate cell-cell adhesions. We have added following text to the manuscript.

“Moreover, the average cell spreading area of the opto-E-cad-MDA cells growing in clusters in the dark was significantly smaller than single cells in the culture (Fig. 4c). In comparison, MDA-MB-231 cells had a similar spreading area to single opto-Ecad-MDA cells and E-cad-MDA cells (MDA-MB-231 cells transfected with E-cadherin (E-cad-MDA) used as a positive control) had a comparable spreading area to opto-E-cad-MDA cells growing in clusters.”

Comment:

3. Biological application: I am completely missing the molecular characterization of adherens junctions. In figure 4, it is shown IFF of actin and p120, but cell densities are rather different when comparing dark and blue light. Also, the WB signal of p120 is reduced under blue light, I do not understand how it is possible that the total protein level changes so quickly depending on the illumination. The cytoplasmic pool of p120 should be the same in the cell lysates, and only the amount localized at AJ should vary. The results on collective migration shown in Fig 5a seem again to suffer from having a different number of cells confined for the wound healing experiment (cells in the dark are more dense than cells exposed to blue light). This affects significantly the collective migration behavior of cell collectives.

There are 2 points raised here by reviewer 1:

1) that cell density is different in conditions in the dark or after blue illumination in the experiments of Immunofluorescence (fig 4B) and collective migration (Fig 5a), which could affect the result. It is true, like reviewer 1 says, that cell density has a strong effect on the organization of cell junctions and on cell migration. In that respect, the authors' response is not fully satisfactory. In particular, they claim that total cell numbers are the same in the dark or light samples, but that is another experiment when cells are far from confluency (Supp. Fig 3a), and does not solve the fact that comparison of adherens junctions and migration as presented in Figures 3 and 4 could result from differences in cell densities.

The authors claim that these observed differences in density are due to different cell spreading and clustering in the dark or under blue light illumination. This could be circumvented by growing the cells in dark conditions until confluency and then illuminating them for several hours.

Response:

Following the suggestion, we repeated the immunofluorescence experiments with cultures where the cells were seeded at confluency. Confirming our previous finding, we observe differences in p120 localization in opto-E-cad cells in the dark and under blue light (Fig. 4b).

For the migration assays (Fig. 5), the experiments were already conducted as suggested. In detail, the cells were grown as a confluent layer in the dark until the wounding. After wounding, we washed the cells and let them equilibrate for 1 hour under dark or blue light, respectively, before acquiring images over 16 hours. This 1-hour step is necessary for stable imaging overnight. For this reason, the appearance at the beginning is different. Therefore, we exclude the possibility that there are effects of cell density. The similar densities are also visible in the provided supplementary movies of the wound healing assay in the dark and under blue light.

We now also quantified the average spreading area of opto-E-cad-MDA cells growing in clusters and as single cells in subconfluent culture and show that cells growing in clusters are on average smaller than single cells.

Comment:

2) that it is surprising that p120 expression decreases so quickly after blue light illumination (Fig. 4d). The authors clarify the experimental conditions, explaining that illumination has been conducted overnight, a timescale compatible with p120 down-regulation. This should be made clearer in the Material and Methods section and in the caption of Fig. 4. In addition, they argue that not only membrane localization but also expression of p120 is affected by E-cadherin expression, and cite several articles to substantiate it. However, it is worth noting that E-cadherin expression is not changed in their experiments after blue light illumination as shown by Fig. 4d, only its adhesive engagement is. How engagement of E-cadherin in adhesive bonds alters p120 stability should be clarified.

Response:

The overnight incubation in the dark or under blue light for data in Fig. 4 is clearly stated in the materials and methods section. Further, we also specify this detail now during the presentation of the data in the main text.

“Yet upon blue illumination overnight, the p120 signal was lower in the cells and the staining was sparser. Likewise, in 3D cellular aggregates of opto-E-cad-MDA cells formed in the dark, the p120 staining was also clearly visible at the cell-cell junctions (Fig. 4d)”

“Complementarily, we observed the expression of opto-E-cad in the opto-E-cad-MDA cells both in the dark and under blue light overnight in western blots (Fig. 4e).”

The interaction between p120 and E-cad is known to be mechanosensitive (Iyer et al., 2019, Current Biology 29, 578–591; Venhuizen *et al.*, 2020, Seminars in Cancer Biology 60, 107-120). Therefore, despite the opto-E-cad expression levels being similar in the dark and under blue light, the interactions with p120 is unlikely to be so.

“The light dependent catenin p120 levels and localization are directly associated with the light-regulated cell-cell adhesions, and they support the molecular picture that the mechanosensitive binding of p120 to the intracellular tail of opto-E-cad and is upregulated in the dark, while under blue light the interaction with the intracellular tail of opto-E-cad is lost and p120 is degraded.”

REVIEWERS' COMMENTS

Reviewer #4 (Remarks to the Author):

Reviewer #4 (Remarks to the Author)

The authors have successfully responded to all my previous comments and interrogations. The added experimental results confirm the ability of opto-E-cadherin to induce functional cell-cell adhesion that are controllable by light in a reversible manner.

I have no more comments and therefore recommend the publication of this article.